# Fast and flexible estimation of effective migration surfaces

**Joseph Marcus[1†]\*, Wooseok Ha[2†]\*, Rina Foygel Barber[3‡]\*, John Novembre[1,4‡]\***

[1]Department of Human Genetics, University of Chicago, Chicago, United States; [2]Department of Statistics, University of California, Berkeley, Berkeley, United States; [3]Department of Statistics, University of Chicago, Chicago, United States; [4]Department of Ecology and Evolution, University of Chicago, Chicago, United States

**Abstract** Spatial population genetic data often exhibits 'isolation-by-distance,' where genetic similarity tends to decrease as individuals become more geographically distant. The rate at which genetic similarity decays with distance is often spatially heterogeneous due to variable population processes like genetic drift, gene flow, and natural selection. Petkova et al., 2016 developed a statistical method called Estimating Effective Migration Surfaces (EEMS) for visualizing spatially heterogeneous isolation-by-distance on a geographic map. While EEMS is a powerful tool for depicting spatial population structure, it can suffer from slow runtimes. Here, we develop a related method called Fast Estimation of Effective Migration Surfaces (FEEMS). FEEMS uses a Gaussian Markov Random Field model in a penalized likelihood framework that allows for efficient optimization and output of effective migration surfaces. Further, the efficient optimization facilitates the inference of migration parameters per edge in the graph, rather than per node (as in EEMS). With simulations, we show conditions under which FEEMS can accurately recover effective migration surfaces with complex gene-flow histories, including those with anisotropy. We apply FEEMS to population genetic data from North American gray wolves and show it performs favorably in comparison to EEMS, with solutions obtained orders of magnitude faster. Overall, FEEMS expands the ability of users to quickly visualize and interpret spatial structure in their data.

**\*For correspondence:**
jhmarcus@uchicago.edu (JM);
haywse@berkeley.edu (WH);
rina@uchicago.edu (RFB);
jnovembre@uchicago.edu (JN)

[†]These authors contributed equally to this work
[‡]These authors also contributed equally to this work

**Competing interests:** The authors declare that no competing interests exist.

## Introduction

The relationship between geography and genetics has had enduring importance in evolutionary biology (see *Felsenstein, 1982*). One fundamental consideration is that individuals who live near one another tend to be more genetically similar than those who live far apart (*Wright, 1943*; *Wright, 1946*; *Malécot, 1948*; *Kimura, 1953*; *Kimura and Weiss, 1964*). This phenomenon is often referred to as 'isolation-by-distance' (IBD) and has been shown to be a pervasive feature in spatial population genetic data across many species (*Slatkin, 1985*; *Dobzhansky and Wright, 1943*; *Meirmans, 2012*). Statistical methods that use both measures of genetic variation and geographic coordinates to understand patterns of IBD have been widely applied (*Bradburd and Ralph, 2019*; *Battey et al., 2020*). One major challenge in these approaches is that the relationship between geography and genetics can be complex. Particularly, geographic features can influence migration in localized regions leading to spatially heterogeneous patterns of IBD (*Bradburd and Ralph, 2019*).

Multiple approaches have been introduced to model spatially non-homogeneous IBD in population genetic data (*McRae, 2006*; *Duforet-Frebourg and Blum, 2014*; *Hanks and Hooten, 2013*; *Petkova et al., 2016*; *Bradburd et al., 2018*; *Al-Asadi et al., 2019*; *Safner et al., 2011*; *Ringbauer et al., 2018*). Particularly relevant to our proposed approach is the work of *Petkova et al., 2016* and *Hanks and Hooten, 2013*. Both approaches model genetic distance using the 'resistance distance' on a weighted graph. This distance metric is inspired by concepts of

effective resistance in circuit theory models, or alternatively understood as the commute time of a random walk on a weighted graph or as a Gaussian graphical model (specifically a conditional auto-regressive process) (*Chandra et al., 1996*; *Hanks and Hooten, 2013*; *Rue and Held, 2005*). Additionally, the resistance distance approach is a computationally convenient and accurate approximation to spatial coalescent models (*McRae, 2006*), although it has limitations in asymmetric migration settings (*Lundgren and Ralph, 2019*).

*Hanks and Hooten, 2013* introduced a Bayesian model that uses measured ecological covariates, such as elevation, to predict genetic distances across sub-populations. Specifically, they use a graph-based model for genotypes observed at different spatial locations. Expected genetic distances across sub-populations in their model are given by resistance distances computed from the edge weights. They parameterize the edge weights of the graph to be a function of known biogeographic covariates, linking local geographic features to genetic variation across the landscape.

Concurrently, the Estimating Effective Migration Surfaces (EEMS) method was developed to help interpret and visualize non-homogeneous gene-flow on a geographic map (*Petkova, 2013*; *Petkova et al., 2016*). EEMS uses resistance distances to approximate the between-sub-population component of pairwise coalescent times in a 'stepping-stone' model of migration and genetic drift (*Kimura, 1953*; *Kimura and Weiss, 1964*). EEMS models the within-sub-population component of pairwise coalescent times, with a node-specific parameter. Instead of using known biogeographic covariates to connect geographic features to genetic variation as in *Hanks and Hooten, 2013*, EEMS infers a set of edge weights (and diversity parameters) that explain the genetic distance data. The inference is based on a hierarchical Bayesian model and a Voronoi-tessellation-based prior to encourage piece-wise constant spatial smoothness in the fitted edge weights.

EEMS uses Markov Chain Monte Carlo (MCMC) and outputs a visualization of the posterior mean for effective migration and a measure of genetic diversity for every spatial position of the focal habitat. Regions with relatively low effective migration can be interpreted to have reduced gene-flow over time, whereas regions with relatively high migration can be interpreted as having elevated gene-flow. EEMS has been applied to multiple systems to describe spatial genetic structure, but despite EEMS's advances in formulating a tractable solution to investigate spatial heterogeneity in IBD, the MCMC algorithm it uses can be slow to converge, in some cases leading to days of computation time for large datasets (*Peter et al., 2020*).

The inference problems faced by EEMS and Hanks and Hooten are related to a growing area referred to as 'graph learning' (*Dong et al., 2019*; *Mateos et al., 2019*). In graph learning, a noisy signal is measured as a scalar value at a set of nodes from the graph, and the aim is then to infer non-negative edge weights that reflect how spatially 'smooth' the signal is with respect to the graph topology (*Kalofolias, 2016*). In population genetic settings, this scalar could be an allele frequency measured at locations in a discrete spatial habitat with effective migration rates between sub-populations. Like the approach taken by *Hanks and Hooten, 2013*, one widely used representation of smooth graph signals is to associate the smoothness property with a Gaussian graphical model where the precision matrix has the form of a graph Laplacian (*Dong et al., 2016*; *Egilmez et al., 2016*). The probabilistic model defined on the graph signal then naturally gives rise to a likelihood for the observed samples, and thus much of the literature in this area focuses on developing specialized algorithms to efficiently solve optimization problems that allow reconstruction of the underlying latent graph. For more information about graph learning and signal processing in general see the survey papers of *Dong et al., 2019* and *Mateos et al., 2019*.

To position the present work in comparison to the 'graph learning' literature, our contributions are twofold. First, in population genetics, it is impossible to collect individual genotypes across all the geographic locations and, as a result, we often work with many, often the majority, of nodes having missing data. As far as we are aware, none of the work in graph signal processing considers this scenario and thus their algorithms are not directly applicable to our setting. In addition, if the number of the observed nodes is much smaller than the number of nodes of a graph, one can project the large matrices associated with the graph to the space of observed nodes, therefore allowing for fast and efficient computation. Second, highly missing nodes in the observed signals can result in significant degradation of the quality of the reconstructed graph unless it is regularized properly. Motivated by the Voronoi-tessellation-based prior adopted in EEMS (*Petkova et al., 2016*), we propose regularization that encourages spatial smoothness in the edge weights.

Building on advances in graph learning, we introduce a method, Fast Estimation of Effective Migration Surfaces (FEEMS), that uses optimization to obtain penalized-likelihood-based estimates of effective migration parameters. In contrast to EEMS which uses a node-specific parameterization of effective migration, we optimize over edge-specific parameters allowing for more flexible migration processes to be fit, such as spatial anisotropy, in which the migration process is not invariant to rotation of the coordinate system (e.g. migration is more extensive along a particular axis). Although we developed this model as a Gaussian Markov Random Field, the resulting likelihood has key similarities to the EEMS model, in that it is a Wishart-distribution that is a function of a genetic distance matrix. Expected genetic distances in both models can be interpreted as 'resistance distances' (*McRae, 2006*).

To fit the model, rather than using MCMC, we develop a fast quasi-Newton optimization algorithm (*Nocedal and Wright, 2006*) and a cross-validation approach for choosing the penalty parameter used in the penalized likelihood. We demonstrate the method using coalescent simulations and an application to a dataset of gray wolves from North America. The output is comparable to the results of EEMS but is provided in orders of magnitude less time. With this improvement in speed, FEEMS opens up the ability to perform fast exploratory data analysis of spatial population structure.

## Results

### Overview of FEEMS

*Figure 1* shows a visual schematic of the FEEMS method. The input data are genotypes and spatial locations (e.g. latitudes and longitudes) for a set of individuals sampled across a geographic region. We construct a dense spatial grid embedded in geographic space where nodes represent sub-populations, and we assign individuals to nodes based on spatial proximity (see *Appendix 1—figure 1* for a visualization of the grid construction and node assignment procedure). The density of the grid is user defined and must be explored to appropriately balance model mis-specification and computational burden. As the density of the lattice increases, the model is similar to discrete approximations used for continuous spatial processes, but the increased density comes at the cost of computational complexity.

Details on the FEEMS model are described in the Materials and methods section, however at a high level, we assume exchangeability of individuals within each sub-population and estimate allele frequencies, $\widehat{f}_j(k)$, for each sub-population, indexed by $k$, and single nucleotide polymorphism (SNP), indexed by $j$, under a simple Binomial sampling model. We also use the recorded sample sizes at each node to model the precision of the estimated allele frequency. With the estimated allele frequencies in hand, we model the data at each SNP using an approximate Gaussian model whose covariance is, up to constant factors, shared across all SNPs—in other words, after rescaling by SNP-specific variation factors, we assume that the set of observed frequencies at each SNP is an independent realization of the same spatial process. The latent frequency variables, $f_j(k)$, are modeled as a Gaussian Markov Random Field (GMRF) with a sparse precision matrix determined by the graph Laplacian and a set of residual variances that vary across SNPs. The pseudo-inverse of the graph Laplacian in a GMRF is inherently connected to the notion of resistance distance in an electrical circuit (*Hanks and Hooten, 2013*) that is often used in population genetics to model the genetic differentiation between sub-populations (*McRae, 2006*). The graph's weighted edges, denoted by $w_{ij}$ between nodes $i$ and $j$, represent gene-flow between the sub-populations (*Friedman et al., 2008*; *Hanks and Hooten, 2013*; *Petkova et al., 2016*). The Gaussian approximation has the advantage that we can analytically marginalize out the latent frequency variables. The resulting likelihood of the observed frequencies shares a number of similarities to that of EEMS (see Materials and methods).

To prevent over-fitting we use penalized maximum likelihood to estimate the edge weights of the graph. Our overall goal is thus to solve the following optimization problem:

$$\widehat{\boldsymbol{w}} = \underset{\boldsymbol{l} \leq \boldsymbol{w} \leq \boldsymbol{u}}{\operatorname{argmin}} \, \ell(\boldsymbol{w}) + \phi_\lambda(\boldsymbol{w}),$$

where $\boldsymbol{w}$ is a vector that stores all the unique elements of the weighted adjacency matrix, $\boldsymbol{l}$ and $\boldsymbol{u}$ are element-wise non-negative lower and upper bounds for $\boldsymbol{w}$, $\ell(\boldsymbol{w})$ is the negative log-likelihood function that comes from the GMRF model described above, and $\phi_\lambda(\boldsymbol{w})$ is a penalty that controls how

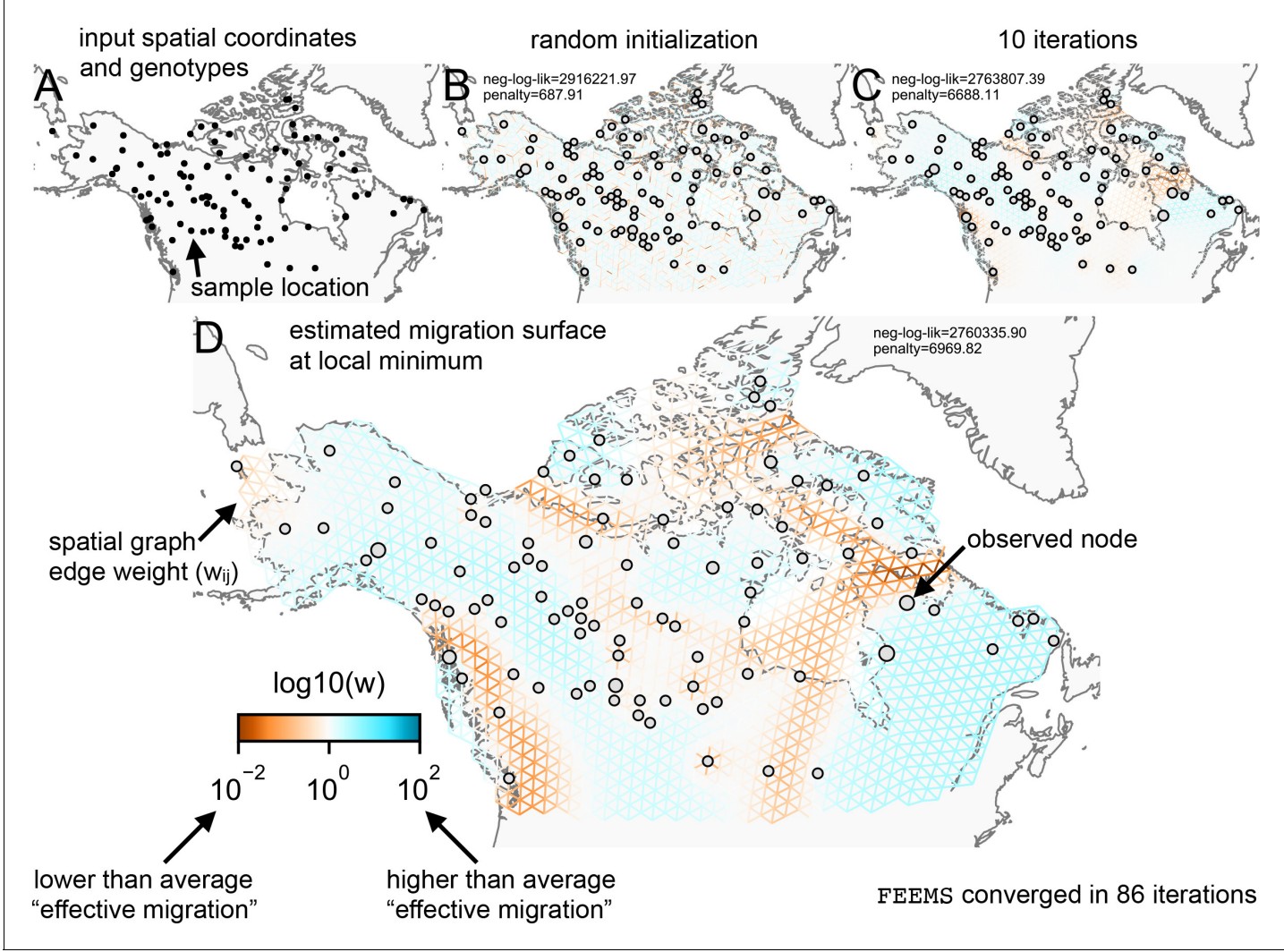

**Figure 1.** Schematic of the FEEMS model: The full panel shows a schematic of going from the raw data (spatial coordinates and genotypes) through optimization of the edge weights, representing effective migration, to convergence of FEEMS to a local optima. (**A**) Map of sample coordinates (black points) from a dataset of gray wolves from North America (*Schweizer et al., 2016*). The input to FEEMS are latitude and longitude coordinates as well as genotype data for each sample. (**B**) The spatial graph edge weights after random initialization uniformly over the graph to begin the optimization algorithm. (**C**) The edge weights after 10 iterations of running FEEMS, when the algorithm has not converged yet. (**D**) The final output of FEEMS after the algorithm has fully converged. The output is annotated with important features of the visualization.

constant or smooth the output migration surface will be and is controlled by the hyperparameter $\lambda > 0$. Writing $\mathcal{V}$ to denote the set of nodes in the graph and $\mathcal{E}(i) \subset \mathcal{V}$ to denote the subset of nodes that have edges connected to node $i$, our penalty is given by

$$\phi_\lambda(\boldsymbol{w}) = \frac{\lambda}{2} \sum_{i \in \mathcal{V}} \sum_{k,\ell \in \mathcal{E}(i)} \left( \log(e^{w_{ik}/\widehat{w_0}} - 1) - \log(e^{w_{i\ell}/\widehat{w_0}} - 1) \right)^2 .$$

This function serves to penalize large differences between the weights $w_{ik}$ and $w_{i\ell}$ on edges that are adjacent, that is, penalizing differences for any pair of edges that share a common node. The tuning parameter $\lambda$ controls the overall strength of the penalization placed on the output of the migration surface—if $\lambda$ is large, the fitted surface will favor a homogeneous set of inferred migration weights on the graph, while if $\lambda$ is low, more flexible graphs can be fitted to recover richer local structure, but this suffers from the potential for over-fitting. The tuning parameter $\lambda$ is selected by evaluating the model's performance at predicting allele frequencies at held out locations using leave-one-out

cross-validation (see Materials and methods '*Leave-one-out cross-validation to select tuning parameters*').

The scale parameter $\widehat{w}_0$ is chosen first fitting a 'constant $w$' model, which is a spatially homogeneous isolation-by-distance model constrained to have a single $w$ value for all edges. In the $\phi_\lambda$ penalty, for adjacent edges $(i, k)$ and $(i, \ell)$, if $w_{ik}$ and $w_{i\ell}$ are large (relative to $\widehat{w}_0$) then the corresponding term of the penalty is approximately proportional to $(w_{ik} - w_{i\ell})^2$, penalizing differences among neighboring edges on a linear scale; if instead $w_{ik}$ and $w_{i\ell}$ are small relative to $\widehat{w}_0$, then the penalty is approximately proportional to $(\log(w_{ik}) - \log(w_{i\ell}))^2$, penalizing differences on a logarithmic scale. In fact, it is also possible to consider treating this scale parameter as a second tuning parameter—we can define a penalty function $\phi_{\lambda,\alpha}(\boldsymbol{w}) = \frac{\lambda}{2} \sum_{i \in \mathcal{V}} \sum_{k,\ell \in \mathcal{E}(i)} (\log(e^{\alpha w_{ik}} - 1) - \log(e^{\alpha w_{i\ell}} - 1))^2$, and explore the solution across different values of both $\lambda$ and $\alpha$. However, we find that empirically choosing $\alpha = 1/\widehat{w}_0$ offers good performance as well as an intuitive interpretation (i.e. scaling edge weights $w_{ik}$ with reference to the constant-$w$ model), and allows us to avoid the computational burden of searching a two-dimensional tuning parameter space.

We use sparse linear algebra routines to efficiently compute the objective function and gradient of our parameters, allowing for the use of widely applied quasi-Newton optimization algorithms (*Nocedal and Wright, 2006*) implemented in standard numerical computing libraries like `scipy` (*Virtanen et al., 2020*) (RRID:SCR_008058). See the Materials and methods section for a detailed description of the statistical models and algorithms used.

## Evaluating FEEMS on 'out of model' coalescent simulations

While our statistical model is not directly based on a population genetic process, it is useful to see how it performs on simulated data under the coalescent stepping stone model (*Figure 2*, also see *Appendix 1—figure 2* for additional scenarios). In these simulations we know, by construction, the model we fit (FEEMS) is different from the true model we simulate data under (the coalescent), allowing us to assess the robustness of the fit to a controlled form of model mis-specification.

The first migration scenario (*Figure 2A–C*) is a spatially homogeneous model where all the migration rates are set to be a constant value on the graph, this is equivalent to simulating data under an homogeneous isolation-by-distance model. In the second migration scenario (*Figure 2D–E*), we simulate a non-homogeneous process by representing a geographic barrier to migration, lowering the migration rates by a factor of 10 in the center of the habitat relative to the left and right regions of the graph. Finally, in the third migration scenario (*Figure 2G–I*), we simulate a pattern which corresponds to anisotropic migration with edges that point east/west being assigned to a fivefold higher migration rate than edges pointing north/south. For each migration scenario, we simulate two sampling designs. In the first 'dense-sampling' design (*Figure 2B,E,I*) we sample individuals for every node of the graph. Next, in the 'sparse-sampling' design (*Figure 2C,F,J*) we sample individuals for only a randomly selected 20% of the nodes.

For each coalescent simulation, we used leave-one-out cross-validation (at the level of sampled nodes) to select the smoothness parameter $\lambda$. In the homogeneous migration simulations, the best value for the smoothness parameter, as determined by the grid value with the lowest leave-one-out cross-validation error, is $\lambda_{\mathrm{cv}} = 100$ in both sampling scenarios with complete and missing data. In the heterogeneous migration simulations $\lambda_{\mathrm{cv}} = 0.298$ with no missing data and $\lambda_{\mathrm{cv}} = 37.927$ with missing data. Finally, in the anisotropic simulations with no missing data $\lambda_{\mathrm{cv}} = 0.298$ and with missing data $\lambda_{\mathrm{cv}} = 0.042$. We note the magnitude of the selected $\lambda$ depends on the scale of the loss function so comparisons across different datasets are not generally interpretable.

With regard to the visualizations of effective migration, FEEMS performs best when all the nodes are sampled on the graph, that is, when there is no missing data (*Figure 2B,E,H*). Interestingly, in the simulated scenarios with many missing nodes, FEEMS can still partly recover the migration history, including the presence of anisotropic migration (*Figure 2I*). A sampling scheme with a central gap leads to a slightly narrower barrier in the heterogeneous migration scenario (*Appendix 1—figure 2I*) and for the anisotropic scenario, a degree of over-smoothness in the northern and southern regions of the center of the graph (*Appendix 1—figure 2N*). For the missing at random sampling design, FEEMS is able to recover the relative edge weights surprisingly well for all scenarios, with the inference being the most challenging when there is anisotropic migration. The potential for FEEMS to recover anisotropic migration is novel relative to EEMS, which was parameterized for

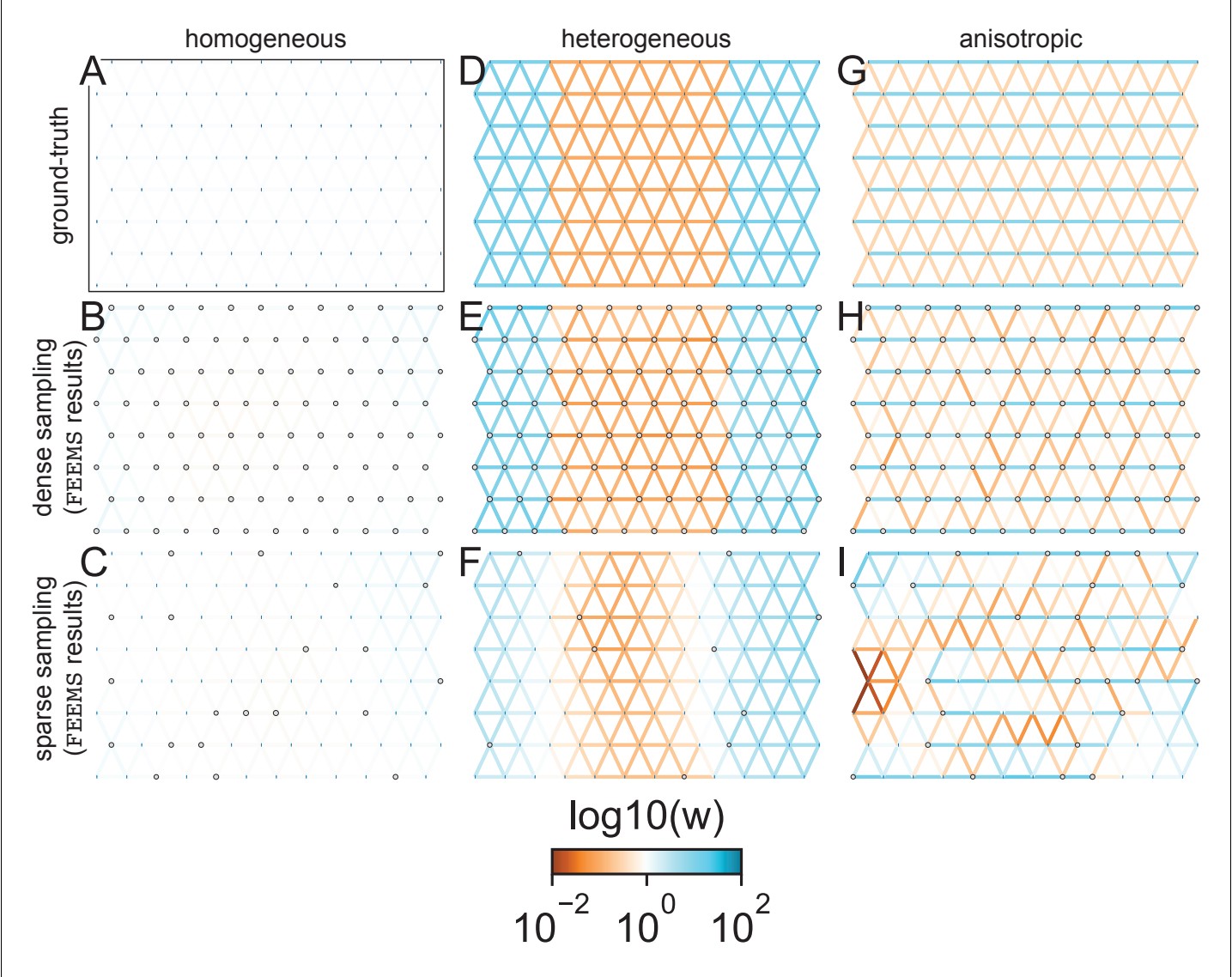

**Figure 2.** FEEMS fit to coalescent simulations: We run FEEMS on coalescent simulations, varying the migration history (columns) and sampling design (rows). In each simulation, we used leave-one-out cross-validation (at the level of sampled nodes) to select the smoothness parameter λ. The first column (A–C) shows the ground-truth and fit of FEEMS to coalescent simulations with a homogeneous migration history that is, a single migration parameter for all edge weights. Note that the ground-truth simulation figures (A,D,G) display coalescent migration rates, not fitted effective migration rates output by FEEMS. The second column (D–F) shows the ground truth and fit of FEEMS to simulations with a heterogeneous migration history that is, reduced gene-flow, with 10-fold lower migration, in the center of the habitat. The third column (G–I) shows the ground truth and fit of FEEMS to an anisotropic migration history with edge weights facing east-west having five fold higher migration than north-south. The second row (B,E,H) shows a sampling design with no missing observations on the graph. The final row (C,F,I) shows a sampling design with 80% of nodes missing at random.

fitting non-stationary isotropic migration histories and produces banding patterns perpendicular to the axis of migration when applied to data from anisotropic coalescent simulations (*Petkova et al., 2016*, Supplementary Figure 2; see also Appendix 1 '*Edge versus node parameterization*' for a related discussion). Overall, even with sparsely sampled graphs, FEEMS is able to produce visualizations that qualitatively capture the migration history in coalescent simulations.

## Application of FEEMS to genotype data from North American gray wolves

To assess the performance of FEEMS on real data, we used a previously published dataset of 111 gray wolves sampled across North America typed at 17,729 SNPs (*Schweizer et al., 2016*;

*Appendix 1—figure 5*). This dataset has a number of advantageous features that make it a useful test case for evaluating FEEMS: (1) The broad sampling range across North America includes a number of relevant geographic features that, a priori, could conceivably lead to restricted gene-flow averaged throughout the population history. These geographic features include mountain ranges, lakes, and islands. (2) The scale of the data is consistent with many studies for non-model systems whose spatial population structure is of interest. For instance, the relatively sparse sampling leads to a challenging statistical problem where there is the potential for many unobserved nodes (sub-populations), depending the density of the grid chosen.

Before applying FEEMS, we confirmed a signature of spatial structure in the data through regressing genetic distances on geographic distances and top genetic PCs against geographic coordinates (*Appendix 1—figures 6*, *7*, *8*, *9*). We also ran multiple replicates of ADMIXTURE for $K = 2$ to $K = 8$, selecting for each $K$ the highest likelihood run among replicates to visualize (*Appendix 1—figure 10*). As expected in a spatial genetic dataset, nearby samples have similar admixture proportions and continuous gradients of changing ancestries are spread throughout the map (*Bradburd et al., 2018*). Whether such gradients in admixture coefficients are due to isolation by distance or specific geographic features that enhance or diminish the levels of genetic differentiation is an interpretive challenge. Explicitly modeling the spatial locations and genetic distance jointly using a method like EEMS or FEEMS is exactly designed to explore these types of questions in the data (*Petkova, 2013*; *Petkova et al., 2016*).

We first show FEEMS results for four different values of the smoothness parameter, λ from large $\lambda = 100$ to small $\lambda = 0.0008$ (*Figure 3*). One interpretation of our regularization penalty is that it encourages fitting models of homogeneous and isotropic migration. When λ is very large (*Figure 3A*), we see FEEMS fits a model where all of the edge weights on the graph nearly equal the mean value, hence all the edge weights are colored white in the relative log-scale. In this case, FEEMS is fitting a relatively homogeneous migration model where all the estimated edge weights get assigned nearly the same value on the graph. As we sequentially lower the penalty parameter, (*Figure 3B,C,D*) the fitted graph begins to appear more complex and heterogeneous as expected (discussed further below). *Figure 3E* shows the cross-validation error for a pre-defined grid of λ values (also see *Appendix 1—figure 6* for visualizations of the fitted versus genetic distance on the full dataset).

The cross-validation approach finds the optimal value of λ to be 2.06. This solution visually appears to have a moderate level of regularization and aligns with several known landscape features (*Figure 4*). Spatial features in the FEEMS visualization qualitatively matches the structure plot output from ADMIXTURE using $K = 6$ (*Appendix 1—figure 10*). We add labels on the figure to highlight a number of pertinent features: (A) St. Lawrence Island, (B) the coastal islands and mountain ranges in British Columbia, (C) the boundary of Boreal Forest and Tundra eco-regions in the Shield Taiga, (D) Queen Elizabeth Islands, (E) Hudson Bay, and (F) Baffin Island. Many of these features were described in *Schweizer et al., 2016* by interpretation of ADMIXTURE, PCA, and $F_{ST}$ statistics. FEEMS is able to succinctly provide an interpretable view of these data in a single visualization. Indeed many of these geographic features plausibly impact gray wolf dispersal and population history (*Schweizer et al., 2016*).

## Comparison to EEMS

We also ran EEMS on the same gray wolf dataset. We used default parameters provided by EEMS but set the number of burn-in iterations to $20 \times 10^6$, MCMC iterations to $50 \times 10^6$, and thinning intervals to 2000. We were unable to run EEMS in a reasonable run time ($\leq 3$ days) for the dense spatial grid of 1207 nodes so we ran EEMS and FEEMS on a sparser graph with 307 nodes.

We find that FEEMS is multiple orders of magnitude faster than EEMS, even when running multiple runs of FEEMS for different regularization settings on both the sparse and dense graphs (*Table 1*). We note that constructing the graph and fitting the model with very low regularization parameters are the most computationally demanding steps in running FEEMS.

We find that many of the same geographic features that have reduced or enhanced gene-flow are concordant between the two methods. The EEMS visualization, qualitatively, best matches solutions of FEEMS with lower λ values (*Figure 4*, *Appendix 1—figure 11*); however, based on the ADMIXTURE results, visual inspection in relation to known geographical features and inspection of

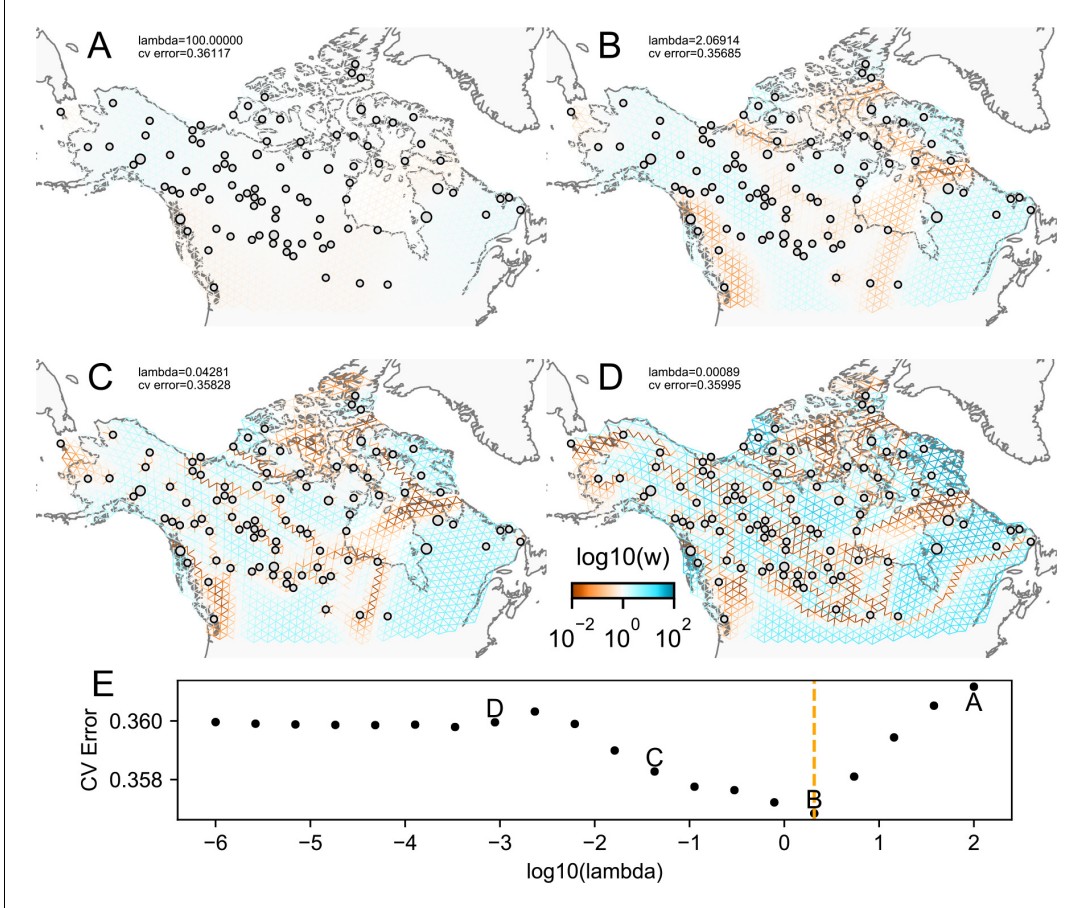

**Figure 3.** The fit of FEEMS to the North American gray wolf dataset for different choices of the smoothing regularization parameter λ: (A) $\lambda = 100$, (B) $\lambda = 2.06$, (C) $\lambda = 0.04$, and (D) $\lambda = 0.0008$. As expected, when λ decreases from large to small (A–D), the fitted graph becomes less smooth and eventually over-fits to the data, revealing a patchy surface in (D), whereas earlier in the regularization path FEEMS fits a homogeneous surface with all edge weights having nearly the same fitted value, as in (A). (E) shows the mean square error between predicted and held-out allele frequencies output by running leave-one-out cross-validation to select the smoothness parameter λ. The cross-validation error is minimized over a pre-selected grid at an intermediate value of $\lambda = 2.06$ as shown in (B).

the observed vs fitted dissimilarity values (*Appendix 1—figures 14*, *22*), we find these solutions to be less satisfying compared to the FEEMS solution found with λ chosen by leave-one-out cross-validation. We note that in many of the EEMS runs the MCMC appears to not have converged (based on visual inspection of trace plots) even after a large number of iterations.

## Discussion

FEEMS is a fast approach that provides an interpretable view of spatial population structure in real datasets and simulations. We want to emphasize that beyond being a fast optimization approach for inferring population structure, our parameterization of the likelihood opens up a number of exciting new directions for improving spatial population genetic inference. Notably, one major difference between EEMS and FEEMS is that in FEEMS each edge weight is assigned its own parameter to be estimated, whereas in EEMS each node is assigned a parameter and each edge is constrained to be the average effective migration between the nodes it connects (see Materials and methods and Appendix 1 '*Edge versus node parameterization*' for details). The node-based parameterization in EEMS makes it difficult to incorporate anisotropy and asymmeteric migration (*Lundgren and Ralph, 2019*). As we have shown here, FEEMS's simple and novel parameterization already has potential to fit anisotropic migration (as shown in coalescent simulations) and may be extendable to other more complex migration processes (such as long-range migration, see below).

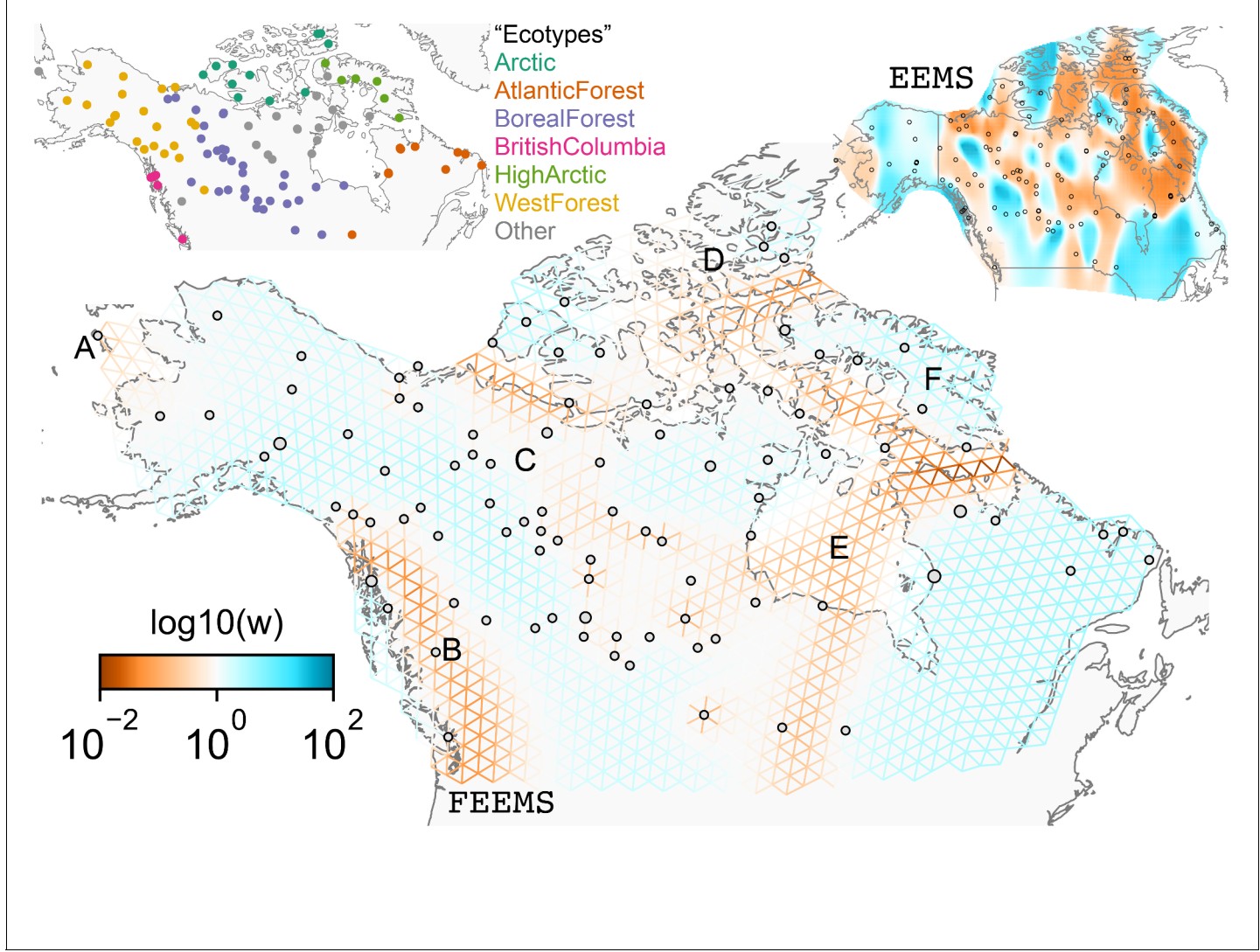

**Figure 4.** FEEMS applied to a population genetic dataset of North American gray wolves: We show the fit of FEEMS applied to a previously published dataset of North American gray wolves. Leave-one-out cross-validation (at the level of sampled nodes) was used to select the smoothness parameter $\lambda = 2.06$. We show the fitted parameters in log-scale with lower effective migration shown in orange and higher effective migration shown in blue. The bold text letters highlights a number of known geographic features that could have plausibly influenced wolf migration over time: (**A**) St. Lawrence Island, (**B**) Coastal mountain ranges in British Columbia, (**C**) The boundary of Boreal Forest and Tundra eco-regions in the Shield Taiga, (**D**) Queen Elizabeth Islands, (**E**) Hudson Bay, and (**F**) Baffin Island. We also display two insets to help interpret the results and compare them to EEMS. In the top left inset we show a map of sample coordinates colored by an ecotype label provided by *Schweizer et al., 2016*. These labels were devised using a combination of genetic and ecological information for 94 'un-admixed' gray wolf samples, and the remaining samples were labeled 'Other'. We can see these ecotype labels align well with the visualization output provided by FEEMS. In the right inset, we display a visualization of the posterior mean effective migration rates from EEMS.

One general challenge, which is not unique to this method, is selecting the tuning parameters controlling the strength of regularization ($\lambda$ in our case). A natural approach is to use cross-validation, which estimates the out-of-sample fit of FEEMS for a particular choice of $\lambda$. We used leave-one-out cross-validation, leaving one sampled population out at a time, and find such an approach works well based on the coalescent simulations and application to the North American wolf data. That said, we sometimes found high variability in the selected solution when we used cross-validation with fewer folds (e.g. five-fold versus leave-one-out, results not shown). We expect this happens when the number of sampled populations is small relative to the complexity of the gene flow landscape, and we recommend using leave-one-out cross-validation in general. We also find it useful to

**Table 1.** Runtimes for FEEMS and EEMS on the North American gray wolf dataset.

We show a table of runtimes for FEEMS and EEMS for two different grid densities, a sparse grid with 307 nodes and a dense grid with 1207 nodes. The second row shows the FEEMS run-times for applying leave-one-out cross-validation to select λ. The third row shows the run-times when applying FEEMS at the best λ value selected using cross-validation. FEEMS is orders of magnitude faster than EEMS, even when using cross-validation to select λ. Runtimes are based on computation using Intel Xeon E5-2680v4 2.4 GHz CPUs with 5 Gb RAM reserved using the University of Chicago Midway2 cluster.

| Method | Sparse grid (run-time) | Dense grid (run-time) |
|---|---|---|
| EEMS | 27.43 hr | N/A |
| FEEMS (Cross-validation) | 10 min 32 s | 1.03 hr |
| FEEMS (Best λ) | 1.23 s | 4.08 s |

fit FEEMS to a sequential grid of regularization parameters and to look at what features are consistent or vary across multiple fits. Informally, one can gain an indication of the strongest features in the data by looking at the order they appear in the regularization path that is, what features overcome the strong penalization of smoothness in the data and that are highly supported by the likelihood. For example, early in the regularization path, we see regions of reduced gene-flow occurring in the west coast of Canada that presumably correspond to Coastal mountain ranges and islands in British Columbia (*Figure 3B*) and this reduced gene-flow appears throughout more flexible fits with lower λ.

An important caveat is that the objective function we optimize is non-convex so any visualization output by FEEMS should be considered a local optimum and, keeping in mind that with different initialization one could get different results. That said, for the datasets investigated, we found the output visualizations were not sensitive to initialization, and thus our default setting is constant initialization fitted under an homogeneous isolation by distance model (See Materials and methods).

When comparing to EEMS, we found FEEMS to be much faster (*Table 1*). While this is encouraging, care must be taken because the goals and outputs of FEEMS and EEMS have a number of differences. FEEMS fits a sequential grid of solutions for different regularization parameters, whereas EEMS infers a posterior distribution and outputs the posterior mean as a point estimate. FEEMS is not a Bayesian method and unlike EEMS, which explores the entire landscape of the posterior distribution, FEEMS returns a particular point estimate: a local minimum point of the optimization landscape. Setting the prior hyper-parameters in EEMS act somewhat like a choice of the tuning parameter λ, except that EEMS uses hierarchical priors that in principle allow for exploration of multiple scales of spatial structure in a single run, but requires potentially long computation times for adequate MCMC convergence.

Like EEMS, FEEMS is based on an assumed underlying spatial graph of populations exchanging gene flow with neighboring populations. While the inferred migration rates explain the data under an assumed model, it is important for users and readers of FEEMS results to keep in mind the range and density of the chosen grid when interpreting results. We note that using a denser grid has the two potential advantages of providing improved approximation for continuously distributed species, as well as a more flexible model space to fit the data.

Depending on the scale of the analysis and the life history of the species, the process of assuming and assigning a single geographic location for each individual is a potential limitation of the modeling framework used here. For instance, the North American wolves studied here are understood to be generally territorial with individual ranges that are on the scale of $10^3$ km$^2$ (*Burch et al., 2005*), which is small relative to the greater than $10^6$ km$^2$ scale of our analysis. Thus, modeling individual wolves with single locations may not generally be problematic. However, at the boundary of the Boreal forest and Tundra, there are wolves with larger annual ranges and seasonal migrations that track caribou herds roughly north-south over distances of 1000 km (*Musiani et al., 2007*), and the wolves in the study were sampled in the winter (*Musiani et al., 2007*; *Schweizer et al., 2016*). If the samples were instead obtained in the summer, the position of the inferred low migration feature near the boundary of the Boreal Forest (marked 'C' in *Figure 4*) would presumably shift northward.

The general cautionary lesson is that one must be careful when interpreting these maps to consider the life history of dispersal for the organism under study during the interpretation of results. Extending the methodology to incorporate knowledge of uncertainty in position or known dispersal may be an interesting direction for future work.

One natural extension to FEEMS, pertinent to a number of biological systems, is incorporating long-range migration (*Pickrell and Pritchard, 2012*; *Bradburd et al., 2016*). In this work, we have used a triangular lattice embedded in geographic space and enforced smoothness in nearby edge weights through penalizing their squared differences (see Materials and methods). We could imagine changing the structure of the graph by adding edges to allow for long-range connections; however, our current regularization scheme would not be appropriate for this setting. Instead, we could imagine adding an additional penalty to the objective, which would only allow a few long range connections to be tolerated. This could be considered to be a combination of two existing approaches for graph-based inference, graphical lasso (GLASSO) and graph Laplacian smoothing, combining the smoothness assumption for nearby connections and the sparsity assumption for long-range connections (*Friedman et al., 2008*; *Wang et al., 2016*). Another potential methodological avenue to incorporate long-range migration is to use a 'greedy' approach. We could imagine adding long-range edges one a time, guided by re-fitting the spatial model and taking a data-driven approach to select particular long-range edges to include. The proposed greedy approach could be considered to be a spatial graph analog of TreeMix (*Pickrell and Pritchard, 2012*).

Another interesting extension would be to incorporate asymmetric migration into the framework of resistance distance and Gaussian Markov Random Field based models. FEEMS, like EEMS, used a likelihood that is based on resistance distances, which are limited in their ability to model asymmetric migration (*Lundgren and Ralph, 2019*). Recently, *Hanks, 2015* developed a promising new framework for deriving the stationary distribution of a continuous time stochastic process with asymmetric migration on a spatial graph. Interestingly, the expected distance of this process has similarities to the resistance distance-based models, in that it depends on the pseudo-inverse of a function of the graph Laplacian. *Hanks, 2015* used MCMC to estimate the effect of known covariates on the edge weights of the spatial graph. Future work could adapt this framework into the penalized optimization approach we have considered here, where adjacent edge weights are encouraged to be smooth.

Finally, when interpreted as mechanistic rather than statistical models, both EEMS and FEEMS implicitly assume time-stationarity, so the estimated migration parameters should be considered to be 'effective' in the sense of being averaged over time in a reality where migration rates are dynamic and changing (*Pickrell and Reich, 2014*). The MAPS method is one recent advance that utilizes long stretches of shared haplotypes between pairs of individuals to perform Bayesian inference of time varying migration rates and population sizes (*Al-Asadi et al., 2019*). With the growing ability to extract high quality DNA from ancient samples, another exciting future direction would be to apply FEEMS to ancient DNA datasets over different time transects in the same focal geographic region to elucidate changing migration histories (*Mathieson et al., 2018*). There are a number of technical challenges in ancient DNA data that make this a difficult problem, particularly high levels of missing and low-coverage data. Our modeling approach could be potentially more robust, in that it takes allele frequencies as input, which may be estimable from dozens of ancient samples at the same spatial location, in spite of high degrees of missingness (*Korneliussen et al., 2014*).

In closing, we look back to a review titled 'How Can We Infer Geography and History from Gene Frequencies?' published in 1982 (*Felsenstein, 1982*). In this review, Felsenstein laid out fundamental open problems in statistical inference in population genetic data, a few of which we restate as they are particularly motivating for our work:

- *For any given covariance matrix, is there a corresponding migration matrix which would be expected to lead to it? If so, how can we find it?*
- *How can we characterize the set of possible migration matrices which are compatible with a given set of observed covariances?*
- *How can we confine our attention to migration patterns which are consistent with the known geometric co-ordinates of the populations?*
- How can we make valid statistical estimates of parameters of stepping stone models?

The methods developed here aim to help address these longstanding problems in statistical population genetics and to provide a foundation for future work to elucidate the role of geography and dispersal in ecological and evolutionary processes.

## Materials and methods

### Model description

See Appendix 1 '*Mathematical notation*' for a detailed description of the notation used to describe the model. To visualize and model spatial patterns in a given population genetic dataset, FEEMS uses an undirected graph, $\mathcal{G} = (\mathcal{V}, \mathcal{E})$ with $|\mathcal{V}| = d$, where nodes represent sub-populations and edge weights $(w_{k\ell})_{(k,\ell) \in \mathcal{E}}$ represent the level of gene-flow between sub-populations $k$ and $\ell$. For computational convenience, we assume $\mathcal{G}$ is a highly sparse graph, specifically a triangular grid that is embedded in geographic space around the sample coordinates. We observe a genotype matrix, $\boldsymbol{Y} \in \mathbb{R}^{n \times p}$, with $n$ rows representing individuals and $p$ columns representing SNPs. We imagine diploid individuals are sampled on the nodes of $\mathcal{G}$ so that $y_{ij}(k) \in \{0, 1, 2\}$ records the count of some arbitrarily predefined allele in individual $i$, SNP $j$, on node $k \in \mathcal{V}$. We assume a commonly used simple Binomial sampling model for the genotypes:

$$y_{ij}(k)\,|\,f_j(k) \sim \mathrm{Binomial}\big(2, f_j(k)\big), \tag{1}$$

where conditional on $f_j(k)$ for all $j, k$, the $y_{ij}(k)$'s are independent. We then estimate an allele frequency at each node and SNP by maximum likelihood:

$$\widehat{f}_j(k) = \frac{\sum_{i=1}^{n_k} y_{ij}(k)}{2n_k},$$

where $n_k$ is the number of individuals sampled at node $k$. We estimate allele frequencies at $o$ of the observed nodes out of $d$ total nodes on the graph. From *Equation (1)*, the estimated frequency in a particular sub-population, conditional on the latent allele frequency, will approximately follow a Gaussian distribution:

$$\widehat{f}_j(k)\,|\,f_j(k) \sim \mathcal{N}\left(f_j(k), \frac{f_j(k)\big(1 - f_j(k)\big)}{2n_k}\right).$$

Using vector notation, we represent the joint model of estimated allele frequencies as:

$$\widehat{\boldsymbol{f}}_j\,|\,\boldsymbol{f}_j \sim \mathcal{N}_o\big(\boldsymbol{A}\boldsymbol{f}_j, \mathrm{diag}(\boldsymbol{d_{f,n}})\big), \tag{2}$$

where $\widehat{\boldsymbol{f}}_j$ is a $o \times 1$ vector of estimated allele frequencies at observed nodes, $\boldsymbol{f}_j$ is a $d \times 1$ vector of latent allele frequencies at all the nodes (both observed and unobserved), and $\boldsymbol{A}$ is a $o \times d$ node assignment matrix where $A_{k\ell} = 1$ if the $k$th estimated allele frequency comes from sub-population $\ell$ and $A_{k\ell} = 0$ otherwise; and $\mathrm{diag}(\boldsymbol{d_{f,n}})$ denotes a $o \times o$ diagonal matrix whose diagonal elements corresponds to the appropriate variance term at observed nodes.

To summarize, we estimate allele frequencies from a subset of nodes on the graph and define latent allele frequencies for all the nodes of the graph. The assignment matrix $\boldsymbol{A}$ maps these latent allele frequencies to our observations. Our summary statistics (the data) are thus $(\widehat{\boldsymbol{F}}, \boldsymbol{n})$ where $\widehat{\boldsymbol{F}}$ is a $o \times p$ matrix of estimated allele frequencies and $\boldsymbol{n}$ is a $o \times 1$ vector of sample sizes for every observed node. We assume the latent allele frequencies come from a Gaussian Markov Random Field:

$$\boldsymbol{f}_j \sim \mathcal{N}_d\big(\mu_j \boldsymbol{I}, \mu_j(1 - \mu_j)\boldsymbol{L}^\dagger\big), \tag{3}$$

where $\boldsymbol{L}$ is the graph Laplacian, $\dagger$ represents the pseudo-inverse operator, and $\mu_j$ represents the average allele frequency across all of the sub-populations. Note that the multiplication by the SNP-specific factor $\mu_j(1 - \mu_j)$ ensures that the variance of the latent allele frequencies vanishes as the average allele frequency approaches to 0 or 1. One interpretation of this model is that the expected squared Euclidean distance between latent allele frequencies on the graph, after being re-scaled by

$\mu_j(1 - \mu_j)$, is exactly the resistance distance of an electrical circuit (**McRae, 2006**; **Hanks and Hooten, 2013**):

$$\frac{\mathbb{E}\left[(f_j(k) - f_j(\ell))^2\right]}{\mu_j(1 - \mu_j)} = r_{k\ell}, \quad \text{where } r_{k\ell} = (\boldsymbol{o}_k - \boldsymbol{o}_\ell)^\top \boldsymbol{L}^\dagger (\boldsymbol{o}_k - \boldsymbol{o}_\ell) = L_{kk}^\dagger - 2L_{k\ell}^\dagger + L_{\ell\ell}^\dagger,$$

where $\boldsymbol{o}_i$ is a one-hot vector (i.e. storing a 1 in element $i$ and zeros elsewhere). It is known that the resistance distance $r_{k\ell}$ is equivalent to the expected commute time between nodes $k$ and $\ell$ of a random walker on the weighted graph $\mathcal{G}$ (**Chandra et al., 1996**). Additionally, the model (**Equation 3**) forms a Markov random field, and thus any latent allele frequency $f_j(k)$ is conditionally independent of all other allele frequencies given its neighbors which are encoded by nonzero elements of $\boldsymbol{L}$ (**Lauritzen, 1996**; **Koller and Friedman, 2009**). Since we use a triangular grid embedded in geographic space to define the graph $\mathcal{G}$, the pattern of nonzero elements is prefixed by the structure of the sparse traingular grid.

Using the law of total variance formula, we can derive from (**Equations 2, 3**) an analytic form for the marginal likelihood. Before proceeding, however, we further approximate the model by assuming $\frac{1}{2}f_j(k)(1 - f_j(k)) \approx \sigma^2 \mu_j(1 - \mu_j)$ for all $j$ and $k$ (see Appendix 1 '*Estimating the edge weights under the exact likelihood model*' for the data model without this approximation). This assumption is mainly for computational purposes and may be a coarse approximation in general. On the other hand, the assumption is not too strong if we exclude SNPs with extremely rare allele frequencies, and more importantly, we find it leads to a good empirical performance, both statistically and computationally. With this approximation, the residual variance parameter $\sigma^2$ is still unknown and needs to be estimated.

Under (**Equation 2, 3**), the law of total variance formula leads to specific formulas for the mean and variance structure as given in (**Equation 4**). With those results, we arrive at the following approximate marginal likelihood:

$$\widehat{\boldsymbol{f}}_j \sim \mathcal{N}_o\left(\mu_j \boldsymbol{I}, \mu_j(1 - \mu_j) \cdot \left(\boldsymbol{A}\boldsymbol{L}^\dagger \boldsymbol{A}^\top + \sigma^2 \mathrm{diag}(\boldsymbol{n}^{-1})\right)\right), \tag{4}$$

where $\mathrm{diag}(\boldsymbol{n}^{-1})$ is a $o \times o$ diagonal matrix computed from the sample sizes at observed nodes. We note the marginal distribution of $\widehat{f}_j$ is not necessarily a Gaussian distribution; however, we use a Gaussian approximation to facilitate computation.

To remove the SNP means we transform the estimated frequencies by a contrast matrix, $\boldsymbol{C} \in \mathbb{R}^{(o-1) \times o}$, that is orthogonal to the one-vector:

$$\boldsymbol{C}\widehat{\boldsymbol{f}}_j \sim \sqrt{\mu_j(1 - \mu_j)} \cdot \mathcal{N}_{o-1}\left(\boldsymbol{0}, \boldsymbol{C}\boldsymbol{A}\boldsymbol{L}^\dagger \boldsymbol{A}^\top \boldsymbol{C}^\top + \sigma^2 \boldsymbol{C}\mathrm{diag}(\boldsymbol{n}^{-1})\boldsymbol{C}^\top\right). \tag{5}$$

Let $\widehat{\boldsymbol{\Sigma}} = \frac{1}{p}\widehat{\boldsymbol{F}}_{\mathrm{s}}\widehat{\boldsymbol{F}}_{\mathrm{s}}^\top$ be the $o \times o$ sample covariance matrix of estimated allele frequencies after re-scaling, that is, $\widehat{\boldsymbol{F}}_{\mathrm{s}}$ is a matrix formed by rescaling the columns of $\widehat{\boldsymbol{F}}$ by $\sqrt{\widehat{\mu}_j(1 - \widehat{\mu}_j)}$, where $\widehat{\mu}_j$ is an estimate of the average allele frequency (see above). We can then express the model in terms of the transformed sample covariance matrix:

$$p \cdot \boldsymbol{C}\widehat{\boldsymbol{\Sigma}}\boldsymbol{C}^\top \sim \mathcal{W}_{o-1}\left(\boldsymbol{C}\boldsymbol{A}\boldsymbol{L}^\dagger \boldsymbol{A}^\top \boldsymbol{C}^\top + \sigma^2 \boldsymbol{C}\mathrm{diag}(\boldsymbol{n}^{-1})\boldsymbol{C}^\top, p\right), \tag{6}$$

where $\mathcal{W}_p$ denotes a Wishart distribution with $p$ degrees of freedom. Note we can equivalently use the sample squared Euclidean distance (often refereed to as a genetic distance) as a summary statistic: letting $\widehat{\boldsymbol{D}}$ be the genetic distance matrix with $\widehat{\boldsymbol{D}}_{k\ell} = \sum_{j=1}^p (\widehat{f}_j(k) - \widehat{f}_j(\ell))^2 / p \cdot \widehat{\mu}_j(1 - \widehat{\mu}_j)$, we have

$$\widehat{\boldsymbol{D}} = \boldsymbol{I}\mathrm{diag}(\widehat{\boldsymbol{\Sigma}})^\top + \mathrm{diag}(\widehat{\boldsymbol{\Sigma}})\boldsymbol{I}^\top - 2\widehat{\boldsymbol{\Sigma}},$$

and so

$$\boldsymbol{C}\widehat{\boldsymbol{D}}\boldsymbol{C}^\top = -2\boldsymbol{C}\widehat{\boldsymbol{\Sigma}}\boldsymbol{C}^\top,$$

using the fact that the contrast matrix $\boldsymbol{C}$ is orthogonal to the one-vector. Thus, we can use the same

spatial covariance model implied by the allele frequencies once we project the distances on to the space of contrasts:

$$-\frac{p}{2} \cdot \boldsymbol{C}\widehat{\boldsymbol{D}}\boldsymbol{C}^{\top} \sim \mathcal{W}_{o-1}\left(\boldsymbol{C}\boldsymbol{A}\boldsymbol{L}^{\dagger}\boldsymbol{A}^{\top}\boldsymbol{C}^{\top} + \sigma^2 \boldsymbol{C}\mathrm{diag}(\boldsymbol{n}^{-1})\boldsymbol{C}^{\top}, p\right).$$

Overall, the negative log-likelihood function implied by our spatial model is the following (ignoring constant terms):

$$\ell(\boldsymbol{w}, \sigma^2; \boldsymbol{C}\widehat{\boldsymbol{\Sigma}}\boldsymbol{C}^{\top}) = p \cdot \mathrm{tr}\left(\left(\boldsymbol{C}\boldsymbol{A}\boldsymbol{L}^{\dagger}\boldsymbol{A}^{\top}\boldsymbol{C}^{\top} + \sigma^2 \boldsymbol{C}\mathrm{diag}(\boldsymbol{n}^{-1})\boldsymbol{C}^{\top}\right)^{-1}\boldsymbol{C}\widehat{\boldsymbol{\Sigma}}\boldsymbol{C}^{\top}\right)$$
$$- p \cdot \log\det\left(\boldsymbol{C}\boldsymbol{A}\boldsymbol{L}^{\dagger}\boldsymbol{A}^{\top}\boldsymbol{C}^{\top} + \sigma^2 \boldsymbol{C}\mathrm{diag}(\boldsymbol{n}^{-1})\boldsymbol{C}^{\top}\right)^{-1}, \tag{7}$$

where $\boldsymbol{w} \in \mathbb{R}^m$ is a vectorized form of the non-zero lower-triangular entries of the weighted adjacency matrix $\boldsymbol{W}$ (recall that the graph Laplacian is completely defined by the edge weights, $\boldsymbol{L} = \mathrm{diag}(\boldsymbol{W1}) - \boldsymbol{W}$, so there is an implicit dependency here). Since the graph is a triangular lattice, we only need to consider the non-zero entries to save computational time, that is, not all sub-populations are connected to each other.

We note our model (*Equation 6*) assumes that the $p$ SNPs are independent. This assumption is unlikely to hold when datasets are analyzed with SNPs that statistically covary (linkage disequilibrium). However, we note that the degree of freedom parameter does not affect the point estimate produced by FEEMS because it is treated as a constant term in the log-likelihood function.

One key difference between EEMS (*Petkova et al., 2016*) and FEEMS is how the edge weights are parameterized. In EEMS, each node is given an effective migration parameter $m_k$ for node $k \in \mathcal{V}$ and the edge weight is parameterized as the average between the nodes it connects, that is, $w_{k\ell} = (m_k + m_\ell)/2$ for $(k, \ell) \in \mathcal{E}$. FEEMS, on the other hand, assigns a parameter to every nonzero edge-weight. The former has fewer parameters, with the specific consequence that it only allows isotropy and imposes an additional degree of similarity among edge weights; instead, in the latter, the edge weights are free to vary apart from the regularization imposed by the penalty. See Appendix 1 '*Edge versus node parameterization*' and *Appendix 1—figures 15*, *17* for more details.

## Penalty description

As mentioned previously, we would like to encourage that nearby edge weights on the graph have similar values to each other. This can be performed by penalizing differences between all edges connected to the same node, that is, spatially adjacent edges:

$$\phi_{\lambda,\alpha}(\boldsymbol{w}) = \frac{\lambda}{2} \sum_{i \in \mathcal{V}} \sum_{k,\ell \in \mathcal{E}(i)} \left(\log(e^{\alpha w_{ik}} - 1) - \log(e^{\alpha w_{i\ell}} - 1)\right)^2,$$

where, as before, $\mathcal{E}(i)$ denotes the set of edges that is connected to node $i$. (As mentioned earlier, in practice we choose $\alpha = 1/\widehat{w}_0$, where $\widehat{w}_0$ is the solution for the 'constant-$w$' model, but we use the free parameter $\alpha$ here for full generality.) The function $x \mapsto \log(e^x - 1)$ (on positive values $x \in (0, \infty)$) is approximately equal to $x$, for $x$ much larger than 1, and is approximately equal to $\log(x)$, for $x$ much smaller than 1. This means that our penalty function effectively penalizes differences on the log scale for edges $(i, k)$ and $(i, \ell)$ with very small weights, but penalizes differences on the original non-log scale for edges with large weights. Using a logarithmic-scale penalty for edges with low weights (rather than simply penalizing $(w_{ik} - w_{i\ell})^2$) leads to smooth graphs for small edge values, and thus allow for an additional degree of flexibility across orders of magnitude of edge weights. The penalty parameter, $\lambda$, controls the overall contribution of the penalty to the objective function. It is convenient to write the penalty in matrix-vector form which we will use throughout:

$$\phi_{\lambda,\alpha}(\boldsymbol{w}) = \frac{\lambda}{2} \|\Delta \log(e^{\alpha \boldsymbol{w}} - \boldsymbol{1})\|_2^2, \tag{8}$$

where $\Delta$ is a signed graph incidence matrix derived from a unweighted graph denoting if pairs of edges are connected to the same node. Specifically, in this expression, we treat $\boldsymbol{w}$ as a vector of length $|\mathcal{E}|$ (i.e. the number of edges), and apply the function $w \mapsto \log(e^{\alpha w} - 1)$ entrywise to this vector. For each pair adjacent edges $(i, k)$ and $(i, \ell)$ in the graph, there is a corresponding row of $\Delta$ with the

value $+1$ in the entry corresponding to edge $(i, k)$, a $-1$ in the entry corresponding to edge $(i, \ell)$, and 0's elsewhere.

One might wonder whether it is possible to use the $\ell_1$ norm in the penalty form *Equation (8)* in place of the $\ell_2$ norm. While it is known that the $\ell_1$ norm might increase local adaptivity and better capture the sharp changes of the underlying structure of the latent allele frequencies (e.g. *Wang et al., 2016*), in our case, we found an inferior performance when using the $\ell_1$ norm over the $\ell_2$ norm—in particular, our primary application of interest is the regime of highly missing nodes, that is, $o \ll d$, in which case the global smoothing seems somewhat necessary to encourage stable recovery of the edge weights at regions with sparsely observed nodes (see Appendix 1 '*Smooth penalty with $\ell_1$ norm*'). In addition, adding the penalty $\phi_{\lambda,\alpha}(\boldsymbol{w})$ allows us to implement faster algorithms to solve the optimization problem due to the differentiability of the $\ell_2$ norm, and as a result, it leads to better overall computational savings and a simpler implementation.

### Optimization

Putting *Equation (7)* and *Equation (8)* together, we infer the migration edge weights $\widehat{\boldsymbol{w}}$ by minimizing the following penalized negative log-likelihood function:

$$
\begin{aligned}
\widehat{\boldsymbol{w}} &= \underset{\boldsymbol{l} \leq \boldsymbol{w} \leq \boldsymbol{u}}{\operatorname{argmin}} \, \ell(\boldsymbol{w}, \sigma^2; \boldsymbol{C}\widehat{\boldsymbol{\Sigma}}\boldsymbol{C}^\top) + \phi_{\lambda,\alpha}(\boldsymbol{w}) \\
&= \underset{\boldsymbol{l} \leq \boldsymbol{w} \leq \boldsymbol{u}}{\operatorname{argmin}} \Big[ p \cdot \operatorname{tr}\Big( \big( \boldsymbol{C}\boldsymbol{A}\boldsymbol{L}^\dagger \boldsymbol{A}^\top \boldsymbol{C}^\top + \sigma^2 \boldsymbol{C}\operatorname{diag}(\boldsymbol{n}^{-1})\boldsymbol{C}^\top \big)^{-1} \boldsymbol{C}\widehat{\boldsymbol{\Sigma}}\boldsymbol{C}^\top \Big) \\
&\qquad - p \cdot \log \det \big( \boldsymbol{C}\boldsymbol{A}\boldsymbol{L}^\dagger \boldsymbol{A}^\top \boldsymbol{C}^\top + \sigma^2 \boldsymbol{C}\operatorname{diag}(\boldsymbol{n}^{-1})\boldsymbol{C}^\top \big)^{-1} + \frac{\lambda}{2} \|\Delta \log(e^{\alpha \boldsymbol{w}} - \boldsymbol{1})\|_2^2 \Big],
\end{aligned}
\tag{9}
$$

where $\boldsymbol{l}, \boldsymbol{u} \in \mathbb{R}_+^m$ represent respectively the entrywise lower- and upper bounds on $\boldsymbol{w}$, that is, we constrain the lower- and upper bound of the edge weights to $\boldsymbol{l}$ and $\boldsymbol{u}$ throughout the optimization. When no prior information is available on the range of the edge weights, we often set $\boldsymbol{l} = \boldsymbol{0}$ and $\boldsymbol{u} = +\infty$.

One advantage of the formulation of *Equation 9* is the use of the vector form parameterization $\boldsymbol{w} \in \mathbb{R}_+^m$ of the symmetric weighted adjacency matrix $\boldsymbol{W} \in \mathbb{R}_+^{d \times d}$. In our triangular graph $\mathcal{G} = (\mathcal{V}, \mathcal{E})$, the number of non-zero lower-triangular entries is $m = \mathcal{O}(d) \ll d^2$, so working directly on the space of vector parameterization saves computational cost. In addition, this avoids the symmetry constraint imposed on the adjacency matrix $\boldsymbol{W}$, hence making optimization easier (*Kalofolias, 2016*).

We solve the optimization problem using a constrained quasi-Newton optimization algorithm, specifically L-BFGS implemented in `scipy` (*Byrd et al., 1995*; *Virtanen et al., 2020*) (RRID:SCR_008058). Since our objective *Equation 9* is non-convex, the L-BFGS algorithm is guaranteed to converge only to a local minimum. Even so, we empirically observe that local minima starting from different initial points are qualitatively similar to each other across many datasets. The L-BFGS algorithm requires gradient and objective values as inputs. Note the naive computation of the objective *Equation 9* is computationally prohibitive because inverting the graph Laplacian has complexity $\mathcal{O}(d^3)$. We take advantage of the sparsity of the graph and specific structure of the problem to efficiently compute gradient and objective values. In theory, our implementation has computational complexity of $\mathcal{O}(do + o^3)$ per iteration which, in the setting of $o \ll d$, is substantially smaller than $\mathcal{O}(d^3)$. It is possible to achieve $\mathcal{O}(do + o^3)$ per-iteration complexity by using a solver that is specially designed for a sparse Laplacian system. In our work, we use sparse Cholesky factorization which may slightly slow down the per-iteration complexity (See Appendix Material for the details of the gradient and objective computation).

### Estimating the residual variance and edge weights under the null model

For estimating the residual variance parameter $\sigma^2$, we first estimate it via maximum likelihood assuming homogeneous isolation by distance. This corresponds to the scenario where every edge-weight in the graph is given the exact same unknown parameter value $w_0$. Under this model, we only have two unknown parameters $w_0$ and the residual variance $\sigma^2$. We estimate these two parameters by jointly optimizing the marginal likelihood using a Nelder-Mead algorithm implemented in `scipy` (*Virtanen et al., 2020*) (RRID:SCR_008058). This requires only likelihood computations which are efficient due to the sparse nature of the graph. This optimization routine outputs an estimate of the

residual variance $\widehat{\sigma}^2$ and the null edge weight $\widehat{w}_0$, which can be used to construct $\boldsymbol{W}(\widehat{w}_0)$ and in turn $\boldsymbol{L}(\widehat{w}_0)$.

One strategy we found effective is to fit the model of homogeneous isolation by distance and then fix the estimated residual variance $\widehat{\sigma}^2$ throughout later fits of the more flexible penalized models—See Appendix 1 '*Jointly estimating the residual variance and edge weights*'. Additionally, we find that initializing the edge weights to $\widehat{w}_0$ to be a useful and intuitive strategy to set the initial values for the entries of $w$ to the correct scale.

## Leave-one-out cross-validation to select tuning parameters

FEEMS estimates one set of graph edge weights for each setting of the tuning parameters $\lambda$ and $\alpha$ which control the smoothness of the fitted edge weights. *Figure 3* shows that the estimated migration surfaces vary substantially depending on the particular choices of the tuning parameters, and indeed, due to the large fraction of unobserved nodes, it can highly over-fit the observed data unless regularized accordingly. To address the issue of selecting the tuning parameters, we propose using leave-one-out cross-validation to assess each fitted model's generalization ability at held out locations.

To simplify the notation, we write the model *Equation 4* for the estimated allele frequencies in SNP $j$ as

$$\widehat{f}_j \sim \mathcal{N}_o\left(\boldsymbol{\mu}_j, \boldsymbol{\Sigma}_j\right),$$

where

$$\boldsymbol{\mu}_j = \mu_j \boldsymbol{I} \text{ and } \boldsymbol{\Sigma}_j = \mu_j(1-\mu_j) \cdot \left(\boldsymbol{A}\boldsymbol{L}^\dagger\boldsymbol{A}^\top + \sigma^2 \text{diag}\left(\boldsymbol{n}^{-1}\right)\right). \tag{10}$$

For each fold, we hold out one node from the set of observed nodes in the graph and use the rest of the nodes to fit FEEMS across a sequential grid of regularization parameters. Note that our objective function is non-convex, so the algorithm converges to different local minima for different regularization parameters, even with the same initial value $\widehat{w}_0$. To stabilize the cross-validation procedure, we recommend using a warm start strategy in which one solves the problem for the largest value of regularization parameters first and use this solution to initialize the algorithm at the next largest value of regularization parameters, and so on. Empirically, we find that using warm starts gives far more reliable model selection than with cold starts, where the problems over the sequence of parameters are solved independently with same initial value $\widehat{w}_0$. We suspect that the poor performance of leave-one-out cross-validation without warm starts is attributed to spatial dependency of allele frequencies and the large fraction of unobserved nodes. Without loss of generality, we assume that the last node has been held out. Re-writing the distribution of the observed frequencies according to the split of observed nodes,

$$\widehat{\boldsymbol{f}}_j = \begin{pmatrix} \widehat{\boldsymbol{f}}_j^{\text{tr}} \\ \widehat{f}_j^{\text{val}} \end{pmatrix} \sim \mathcal{N}_o\left( \begin{pmatrix} \boldsymbol{\mu}_j^{\text{tr}} \\ \mu_j^{\text{val}} \end{pmatrix}, \begin{pmatrix} \boldsymbol{\Sigma}_j^{\text{tr}} & \boldsymbol{\Sigma}_j^{\text{cov}} \\ \boldsymbol{\Sigma}_j^{\text{cov}\top} & \Sigma_j^{\text{val}} \end{pmatrix} \right),$$

the conditional mean of the observed frequency $\widehat{f}_j^{\text{val}}$ on the held out node, given the rest, is given by

$$\widehat{f}_j^{\text{val,pred}} = \mathbb{E}\left[\widehat{f}_j^{\text{val}} \mid \widehat{\boldsymbol{f}}_j^{\text{tr}}\right] = \mu_j^{\text{val}} + \boldsymbol{\Sigma}_j^{\text{cov}\top} \boldsymbol{\Sigma}_j^{\text{tr}-1} \left(\widehat{\boldsymbol{f}}_j^{\text{tr}} - \boldsymbol{\mu}_j^{\text{tr}}\right).$$

Using this formula, we can predict allele frequencies at held out locations using the fitted graph $\widehat{\boldsymbol{L}} = \widehat{\boldsymbol{L}}(\lambda, \alpha)$ for each setting of tuning parameters $\lambda$ and $\alpha$. Note that in *Equation (10)*, the parameters $\mu_j$ and $\sigma$ are also unknown, and we use an estimate of the average allele frequency $\widehat{\mu}_j$ and the estimated residual variance $\widehat{\sigma}$ from the 'constant-$w$' model (they are not dependent on $\lambda$ and $\alpha$). Then we select the tuning parameters $\lambda$ and $\alpha$ that output the minimum prediction error averaged over all SNPs $\frac{1}{p}\sum_j \left\|\widehat{f}_j^{\text{val,pred}} - \widehat{f}_j^{\text{val}}\right\|_2^2$, averaged over all the held out nodes (with $o$ observed nodes in total). As mentioned earlier, in practice we choose $\alpha = 1/\widehat{w}_0$ and hence we can use the leave-one-out cross-validation to search for $\lambda$ only, which allows us to avoid the computational cost of searching over the two-dimensional parameter space.

## Comparison between FEEMS and EEMS models

At a high level, we can summarize the differences between FEEMS and EEMS as follows: (1) the likelihood functions of FEEMS and EEMS are slightly different as a function of the graph Laplacian $\boldsymbol{L}$; (2) the migration rates are parameterized in terms of edge weights or in terms of node weights; and (3) EEMS is based on Bayesian inference and thus chooses a prior and studies the posterior distribution, while FEEMS is an optimization-based approach and thus chooses a penalty function and minimizes the penalized log-likelihood (in particular, the EEMS prior and the FEEMS penalty are both aiming for locally constant type migration surfaces). The last two points were already discussed in the above sections, so here we focus on the difference of the likelihoods between the two methods.

FEEMS develops the spatial model for the genetic differentiation through Gaussian Markov Random Field, but the resulting likelihood has similarities to EEMS (*Petkova et al., 2016*) which considers the pairwise coalescent times. Using our notation, we can write the EEMS model as

$$-\frac{\nu}{2} \cdot \boldsymbol{C}\widehat{\boldsymbol{D}}^* \boldsymbol{C}^\top \sim \mathcal{W}_{o-1}\left(\sigma^* \cdot \boldsymbol{C}\left(\frac{1}{2}\boldsymbol{A}\boldsymbol{L}^\dagger \boldsymbol{A}^\top + \mathrm{diag}(\boldsymbol{A}\boldsymbol{q})\right)\boldsymbol{C}^\top, \nu\right), \tag{11}$$

where $\nu \in [o-1, p]$ is the effective degree of freedom, $\sigma^* > 0$ is the scale nuisance parameter, and $\boldsymbol{q}$ is a $d \times 1$ vector of the within-sub-population coalescent rates. $\widehat{\boldsymbol{D}}^*$ represents the genetic distance matrix without re-scaling, where the $(k, \ell)$-th element is given by $\widehat{\boldsymbol{D}}_{k\ell}^* = \sum_{j=1}^p (\widehat{f}_j(k) - \widehat{f}_j(\ell))^2 / p$. That is, unlike FEEMS, EEMS does not consider the SNP-specific re-scaling factor $\mu_j(1-\mu_j)$ to account for the vanishing variance of the observed allele frequencies as the average allele frequency approaches to 0 or 1.

In *Equation (11)*, the effective degree of freedom $\nu$ is introduced to account for the dependency across SNPs in close proximity. Because EEMS uses a hierarchical Bayesian model to infer the effective migration rates, $\nu$ is being estimated alongside other model parameters. On the other hand, FEEMS uses an optimization-based approach and the degrees of freedom has no influence on the point estimate of the migration rates. Besides the effective degree of freedom and the SNP-specific re-scaling by $\mu_j(1-\mu_j)$, the EEMS and FEEMS likelihoods are equivalent up to constant factors, as long as only one individual is observed per node and the residual variance $\sigma^2$ is allowed to vary across nodes—See Appendix 1 '*Jointly estimating the residual variance and edge weights*' for details. The constant factors, such as $\sigma^*$, can be effectively absorbed into the unknown model parameters $\boldsymbol{L}$ and $\boldsymbol{q}$ and therefore they do not affect the estimation of effective migration rates, up to constant factors.

## Data description and quality control

We analyzed a population genetic dataset of North American gray wolves previously published in *Schweizer et al., 2016*. For this, we downloaded plink (RRID:SCR_001757) formatted files and spatial coordinates from https://doi.org/10.5061/dryad.c9b25. We removed all SNPs with minor allele frequency less than 5% and with missingness greater then 10%, resulting in a final set of 111 individuals and 17,729 SNPs.

## Population structure analyses

We fit the Pritchard, Donnelly, and Stephens model (PSD) and ran principal components analysis on the genotype matrix of North American gray wolves (*Price et al., 2006*; *Pritchard et al., 2000*). For the PSD model, we used the ADMIXTURE software (RRID:SCR_001263) on the un-normalized genotypes, running five replicates per choice of $K$, from $K = 2$ to $K = 8$ (*Alexander et al., 2009*). For each $K$, we choose the one that achieved the highest likelihood to visualize. For PCA, we centered and scaled the genotype matrix and then ran sklearn (RRID:SCR_019053) implementation of PCA, truncated to compute 50 eigenvectors.

## Grid construction

To create a dense triangular lattice around the sample locations, we first define an outer boundary polygon. As a default, we construct the lattice by creating a convex hull around the sample points and manually trimming the polygon to adhere to the geography of the study organism and balancing the sample point range with the extent of local geography using the following website https://

www.keene.edu/campus/maps/tool/. We often do not exclude internal 'holes' in the habitat (e.g. water features for terrestrial animals), and let the model instead fit effective migration rates for those features to the extent they lead to elevated differentiation. We also emphasize the importance of defining the lattice for FEEMS as well as EEMS and suggest this should be carefully curated with prior biological knowledge about the system.

To ensure edges cover an equal area over the entire region, we downloaded and intersected a uniform grid defined on the spherical shape of earth (*Sahr et al., 2003*). These defined grids are pre-computed at a number of different resolutions, allowing a user to test FEEMS at different grid densities which is an important feature to explore.

## Code availability

The code to reproduce the results of this paper and more can be found at https://github.com/jhmarcus/feems-analysis (*Marcus and Ha, 2021a*, copy archived at swh:1:rev:f2d7330f25f8a11124d-b09000918ae38ae00d4a7, *Marcus and Ha, 2021b*). A `python` (RRID:SCR_008394) package implementing the method can be found at https://github.com/Novembrelab/feems.

## Acknowledgements

We thank Rena Schweizer for helping us download and process the gray wolf dataset used in the paper, Ben Peter for providing feedback and code for helping to construct the discrete global grids and preparing the human genetic dataset, and Hussein Al-Asadi, Peter Carbonetto, Dan Rice for helpful conversations about the optimization and modeling approach. We also acknowledge helpful feedback from Arjun Biddanda, Anna Di Rienzo, Matthew Stephens, the Stephens Lab, the Novembre Lab, and the University of Chicago 4th floor Cummings Life Science Center computational biology community. This study was supported in part by the National Science Foundation via fellowship DGE-1746045 and the National Institute of General Medical Sciences via training grant T32GM007197 to JHM and R01GM132383 to JN. WH was partially supported by the NSF via the TRIPODS program and by the Berkeley Institute for Data Science. RFB was supported by the National Science Foundation via grant DMS–1654076, and by the Office of Naval Research via grant N00014-20-1-2337.

## Additional information

### Funding

| Funder | Grant reference number | Author |
| --- | --- | --- |
| National Science Foundation | DGE-1746045 | Joseph Marcus |
| National Institute of General Medical Sciences | T32GM007197 | Joseph Marcus |
| National Institute of General Medical Sciences | R01GM132383 | John Novembre |
| National Science Foundation | TRIPODS Program | Wooseok Ha |
| University of California Berkeley | Institute for Data Science | Wooseok Ha |
| National Science Foundation | DMS-1654076 | Rina Foygel Barber |
| Office of Naval Research | N00014-20-1-2337 | Rina Foygel Barber |

The funders had no role in study design, data collection and interpretation, or the decision to submit the work for publication.

### Author contributions

Joseph Marcus, Conceptualization, Software, Formal analysis, Visualization, Methodology, Writing - original draft, Writing - review and editing; Wooseok Ha, Software, Formal analysis, Visualization, Methodology, Writing - original draft, Writing - review and editing; Rina Foygel Barber, Formal

analysis, Supervision, Methodology, Writing - review and editing; John Novembre, Conceptualization, Formal analysis, Supervision, Visualization, Writing - review and editing

**Author ORCIDs**
Joseph Marcus ⬚ https://orcid.org/0000-0002-0923-9881
Wooseok Ha ⬚ https://orcid.org/0000-0002-9069-854X
John Novembre ⬚ https://orcid.org/0000-0001-5345-0214

**Decision letter and Author response**
Decision letter https://doi.org/10.7554/eLife.61927.sa1
Author response https://doi.org/10.7554/eLife.61927.sa2

## Additional files

**Supplementary files**
• Transparent reporting form

**Data availability**
Genotyping data can be found at https://doi.org/10.5061/dryad.c9b25 and stored in the FEEMS python package at https://github.com/Novembrelab/feems (copy archived at https://archive.soft-wareheritage.org/swh:1:rev:2df82f92ba690f5fd98aee6612b155d973ffb12d).

The following previously published dataset was used:

| Author(s) | Year | Dataset title | Dataset URL | Database and Identifier |
|---|---|---|---|---|
| Schweizer RM, vonHoldt JC, R, BM, Harrigan, Knowles, Musiani M, Coltman D, Novembre J, Wayne RK | 2016 | Genetic subdivision and candidate genes under selection in North American grey wolves | https://doi.org/10.5061/dryad.c9b25 | Dryad Digital Repository, 10.5061/dryad.c9b25 |

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

## Appendix 1

### Mathematical notation

We denote matrices using bold capital letters $A$. Bold lowercase letters are vectors $a$, and non-bold lowercase letters are scalars $a$. We denote by $A^{-1}$ and $A^\dagger$ the inverse and (Moore-Penrose) pseudo-inverse of $A$, respectively. We use $y \sim N_p(\mu, \Sigma)$ to express that the random vector $y$ is modeled as a $p$-dimensional multivariate Gaussian distribution with fixed parameters $\mu$ and $\Sigma$ and use the conditional notation $y|\mu \sim N_p(\mu, \Sigma)$ if $\mu$ is random.

A graph is a pair $\mathcal{G} = (\mathcal{V}, \mathcal{E})$, where $\mathcal{V}$ denotes a set of nodes or vertices and $\mathcal{E} \subseteq \mathcal{V} \times \mathcal{V}$ denotes a set of edges. Throughout we assume the graph $\mathcal{G}$ is undirected, weighted, and contains no self loops, that is, $(k, \ell) \in \mathcal{E} \Longleftrightarrow (\ell, k) \in \mathcal{E}$ and $(k, k) \notin \mathcal{E}$ and each edge $(k, \ell) \in \mathcal{E}$ is given a weight $w_{k\ell} = w_{\ell k} > 0$. We write $W$ to indicate the symmetric weighted adjacency matrix, that is,

$$W_{k\ell} = \begin{cases} w_{k\ell}, & \text{if } (k, \ell) \in \mathcal{E}, \\ 0, & \text{otherwise.} \end{cases}$$

$w \in \mathbb{R}^m$ is a vectorized form of the non-zero lower-triangular entries of $W$ where $m = |\mathcal{E}|/2$ is the number of non-zero lower triangular elements. We denote by $L = \mathrm{diag}(W1) - W$ the graph Laplacian.

### Gradient computation

In practice, we make a change of variable from $w \in \mathbb{R}^m_+$ to $z = \log(w) \in \mathbb{R}^m$ and the algorithm is applied to the transformed objective function:

$$\ell(\exp(z), \sigma^2; C\widehat{\Sigma}C^\top) + \phi_{\lambda,\alpha}(\exp(z)) = \widetilde{\ell}(z, \sigma^2; C\widehat{\Sigma}C^\top) + \widetilde{\phi}_{\lambda,\alpha}(z).$$

After the change of variable, the objective value remains the same, whereas it follows from the chain rule that $\nabla(\widetilde{\ell}(z) + \widetilde{\phi}_{\lambda,\alpha}(z)) = \nabla(\ell(w) + \phi_{\lambda,\alpha}(w)) \odot w$ where $\odot$ indicates the Hadamard product or elementwise product—for notational convenience, we drop the dependency of $\ell$ on the quantities $\sigma^2$ and $C\widehat{\Sigma}C^\top$. Furthermore, the computation of $\nabla\phi_{\lambda,\alpha}(w)$ is relatively straightforward, so in the rest of this section, we discuss only the computation of the gradient of the negative log-likelihood function with respect to $w$, that is, $\nabla\ell(w)$.

Recall, by definition, the graph Laplacian $L$ implicitly depends on the variable $w$ through $L = \mathrm{diag}(W1) - W$. Throughout we assume the first $o$ rows and columns of $L$ correspond to the observed nodes. With this assumption, our node assignment matrix has block structure $A = [I_{o \times o} \mid 0_{o \times (d-o)}]$. To simplify some of the equations appearing later, we introduce the notation: we define

$$L_{\text{full}} := L + \frac{11^\top}{d}, \quad \Sigma := AL_{\text{full}}^{-1}A^\top + \sigma^2 \mathrm{diag}(n^{-1}), \tag{12}$$

and

$$M := C^\top \left( (C\Sigma C)^{-1}(C\widehat{\Sigma}C)(C\Sigma C)^{-1} - (C\Sigma C)^{-1} \right) C.$$

Applying the chain rule and matrix derivatives, we can calculate:

$$\nabla\ell(w) = \frac{\partial\ell(w)}{\partial\mathrm{vec}(L)} \cdot \frac{\partial\mathrm{vec}(L)}{\partial w^\top},$$

where vec is the vectorization operator and $\partial\ell/\partial\mathrm{vec}(L)$ and $\partial\mathrm{vec}(L)/\partial w^\top$ are $1 \times d^2$ vector and $d^2 \times d$ matrix, respectively, given by

$$\frac{\partial\ell(w)}{\partial\mathrm{vec}(L)} = p \cdot \mathrm{vec}\left(L_{\text{full}}^{-1}A^\top MAL_{\text{full}}^{-1,\top}\right), \quad \frac{\partial\mathrm{vec}(L)}{\partial w^\top} = S - T. \tag{13}$$

Here, $S$ and $T$ are linear operators that satisfy $Sw = \mathrm{diag}(W1)$ and $Tw = W$. Note $S$ and $T$ both have $\mathcal{O}(d)$ many nonzero entries, so we can perform sparse matrix multiplication to efficiently

compute the matrix-vector multiplication $\partial \ell / \partial \mathrm{vec}(\boldsymbol{L}) \cdot (\boldsymbol{S} - \boldsymbol{T})$. On the other hand, the computation of $\partial \ell / \partial \mathrm{vec}(\boldsymbol{L})$ is more challenging as it requires inverting the full $d \times d$ matrix $\boldsymbol{L}_{\mathrm{full}}$. Next, we develop a procedure that efficiently computes $\partial \ell / \partial \mathrm{vec}(\boldsymbol{L})$. We proceed by dividing the task into multiple steps.

## 1. Computing $\boldsymbol{\Sigma}^{-1}$

Recalling the block structure $A = [\mathbf{I}_{o \times o} \mid \boldsymbol{0}_{o \times (d-o)}]$ of the node assignment matrix, we can write $\boldsymbol{\Sigma}$ as:

$$\boldsymbol{\Sigma} = \left(\boldsymbol{L}_{\mathrm{full}}^{-1}\right)_{o \times o} + \sigma^2 \mathrm{diag}(\boldsymbol{n}^{-1}),$$

where $\left(\boldsymbol{L}_{\mathrm{full}}^{-1}\right)_{o \times o}$ denotes the $o \times o$ upper-left block of $\boldsymbol{L}_{\mathrm{full}}^{-1}$. Following *Petkova et al., 2016*, the inverse $\boldsymbol{\Sigma}^{-1}$ has the form

$$\boldsymbol{\Sigma}^{-1} = X + \sigma^{-2} \mathrm{diag}(\boldsymbol{n}), \tag{14}$$

for some matrix $X \in \mathbb{R}^{o \times o}$. Equating $\boldsymbol{\Sigma}\boldsymbol{\Sigma}^{-1} = \mathbf{I}$, it follows that

$$\left[\left(\boldsymbol{L}_{\mathrm{full}}^{-1}\right)_{o \times o} + \sigma^2 \mathrm{diag}(\boldsymbol{n}^{-1})\right]\left(X + \sigma^{-2}\mathrm{diag}(\boldsymbol{n})\right) = \mathbf{I}$$

$$\Longleftrightarrow \left[\left(\boldsymbol{L}_{\mathrm{full}}^{-1}\right)_{o \times o} + \sigma^2 \mathrm{diag}(\boldsymbol{n}^{-1})\right]X = -\sigma^{-2}\left(\boldsymbol{L}_{\mathrm{full}}^{-1}\right)_{o \times o}\mathrm{diag}(\boldsymbol{n}). \tag{15}$$

Therefore, $\boldsymbol{\Sigma}^{-1}$ can be obtained by solving the $o \times o$ linear system *Equation (15)* and plugging the solution into *Equation (14)*. The challenge here is to compute $\left(\boldsymbol{L}_{\mathrm{full}}^{-1}\right)_{o \times o}$ without matrix inversion of the full-dimensional $\boldsymbol{L}_{\mathrm{full}}$.

## 2. Computing $\left(\boldsymbol{L}_{\mathrm{full}}^{-1}\right)_{o \times o}$

Let $\boldsymbol{L}_{\mathrm{full}, o \times o}$ be the $o \times o$ block matrix corresponding to the observed nodes of $\boldsymbol{L}_{\mathrm{full}}$, and similarly let $\boldsymbol{L}_{\mathrm{full}, (d-o) \times (d-o)}$ and $\boldsymbol{L}_{\mathrm{full}, o \times (d-o)} = \boldsymbol{L}_{\mathrm{full}, (d-o) \times o}^{\top}$ be the corresponding block matrices of $\boldsymbol{L}_{\mathrm{full}}$, respectively. The inverse of $\left(\boldsymbol{L}_{\mathrm{full}}^{-1}\right)_{o \times o}$ is then given by the Schur complement of $\boldsymbol{L}_{\mathrm{full}, (d-o) \times (d-o)}$ in $\boldsymbol{L}$:

$$\left[\left(\boldsymbol{L}_{\mathrm{full}}^{-1}\right)_{o \times o}\right]^{-1} = \boldsymbol{L}_{\mathrm{full}, o \times o} - \boldsymbol{L}_{\mathrm{full}, o \times (d-o)}\left(\boldsymbol{L}_{\mathrm{full}, (d-o) \times (d-o)}\right)^{-1}\boldsymbol{L}_{\mathrm{full}, (d-o) \times o}. \tag{16}$$

See also *Hanks and Hooten, 2013*, *Petkova et al., 2016*. Since every term in *Equation (16)* has sparse + rank-1 structure, the matrix multiplications can be performed fast. In addition, for the term $\left(\boldsymbol{L}_{\mathrm{full}, (d-o) \times (d-o)}\right)^{-1}$, we can use the Sherman-Morrison formula so that the inverse is given explicitly by

$$\begin{aligned}
\left(\boldsymbol{L}_{\mathrm{full}, (d-o) \times (d-o)}\right)^{-1} &= \left(\boldsymbol{L}_{(d-o) \times (d-o)} + \frac{\boldsymbol{1}\boldsymbol{1}^{\top}}{d}\right)^{-1} \\
&= \boldsymbol{L}_{(d-o) \times (d-o)}^{-1} - \frac{1}{d + \boldsymbol{1}^{\top}\boldsymbol{L}_{(d-o) \times (d-o)}^{-1}\boldsymbol{1}}\boldsymbol{L}_{(d-o) \times (d-o)}^{-1}\boldsymbol{1}\boldsymbol{1}^{\top}\boldsymbol{L}_{(d-o) \times (d-o)}^{-1}.
\end{aligned}$$

Hence, in order to compute $\left(\boldsymbol{L}_{\mathrm{full}, (d-o) \times (d-o)}\right)^{-1}\boldsymbol{L}_{\mathrm{full}, (d-o) \times o}$, we need to solve two systems of linear equations:

$$\boldsymbol{L}_{(d-o) \times (d-o)}U = \boldsymbol{L}_{\mathrm{full}, (d-o) \times o} \ \text{ and } \ \boldsymbol{L}_{(d-o) \times (d-o)}\boldsymbol{u} = \boldsymbol{1}.$$

Note that the matrix $\boldsymbol{L}_{(d-o) \times (d-o)}$ is sparse, so both systems can be solved efficiently by performing sparse Cholesky factorization on $\boldsymbol{L}_{(d-o) \times (d-o)}$ (*Hanks and Hooten, 2013*). Alternatively, one can implement fast Laplacian solvers (*Vishnoi, 2013*) that solve the Laplacian system in time nearly linear in the dimension $\mathcal{O}(d)$. After we obtain $\left[\left(\boldsymbol{L}_{\mathrm{full}}^{-1}\right)_{o \times o}\right]^{-1}$ via sparse + rank-1 matrix multiplication and sparse Cholesky factorization, we can invert the $o \times o$ matrix to get $\left(\boldsymbol{L}_{\mathrm{full}}^{-1}\right)_{o \times o}$.

### 3. Computing $\left(L_{\text{full}}^{-1}\right)_{d \times o}$

We write

$$\left(L_{\text{full}}^{-1}\right)_{d \times o} = \begin{bmatrix} \left(L_{\text{full}}^{-1}\right)_{o \times o} \\ \left(L_{\text{full}}^{-1}\right)_{(d-o) \times o} \end{bmatrix}.$$

Using the inversion of the matrix in a block form, the $(d - o) \times o$ block component is given by

$$\left(L_{\text{full}}^{-1}\right)_{(d-o) \times o} = - \underbrace{\left(L_{\text{full},(d-o) \times (d-o)}\right)^{-1} L_{\text{full},(d-o) \times o}}_{(A)} \underbrace{\left(L_{\text{full}}^{-1}\right)_{o \times 0}}_{(B)} \tag{17}$$

Since each of the two terms (A) and (B) has been already computed in the previous step, there is no need to recompute them. In total, it requires a $(d - o) \times o$ matrix and $o \times o$ matrix multiplication.

### 4. Computing the full gradient

Going back to the expression of $\nabla \ell(w)$ in *Equation (13)*, and noting the block structure of the assignment matrix $A$, we have:

$$\frac{\partial \ell(w)}{\partial \text{vec}(L)} = p \cdot \text{vec}\left(\left(L_{\text{full}}^{-1}\right)_{d \times o} M \left(L_{\text{full}}^{-1}\right)_{d \times o}^{\top}\right).$$

Define $\Pi_I = I\left(I^{\top} \Sigma^{-1} I\right)^{-1} I^{\top} \Sigma^{-1}$ which acts as a sort of projection to the space of constant vectors with respect to the inner product $\langle x, y \rangle = x^{\top} \Sigma^{-1} y$. Using the identity $I - \Pi_I = \Sigma C^{\top} (C \Sigma C^{\top})^{-1} C$ (*McCullagh, 2009*), then we can write $M$ in terms of $\Pi_I$:

$$M = \Sigma^{-1} (I - \Pi_I) \widehat{\Sigma} \Sigma^{-1} (I - \Pi_I) - \Sigma^{-1} (I - \Pi_I). \tag{18}$$

Since $\Pi_I$ is a rank-1 matrix, this expression of $M$ allows easier computation. Finally we can put together *Equation (14)*, *Equation (15)*, *Equation (17)*, and *Equation (18)*, to compute the gradient of the negative log-likelihood function with respect to the graph Laplacian.

## Objective computation

The graph Laplacian $L$ is orthogonal to the one vector 1, so using the notation introduced in *Equation (12)*, we can express our objective function as

$$\ell(w) + \phi_{\lambda,\alpha}(w) = p \cdot \text{tr}\left(\left(C \Sigma C^{\top}\right)^{-1} C \widehat{\Sigma} C^{\top}\right) - p \cdot \log \det(C \Sigma C)^{-1} + \frac{\lambda}{2} \|\Delta \log(e^{\alpha w} - I)\|_2^2.$$

With the identity $I - \Pi_I = \Sigma C^{\top} (C \Sigma C^{\top})^{-1} C$, the trace term is:

$$\text{tr}\left(\left(C \Sigma C^{\top}\right)^{-1} C \widehat{\Sigma} C^{\top}\right) = \text{tr}\left(C^{\top} \left(C \Sigma C^{\top}\right)^{-1} C \widehat{\Sigma}\right) = \text{tr}\left(\Sigma^{-1} (I - \Pi_I) \widehat{\Sigma}\right).$$

The matrix inside the trace has been constructed in the gradient computation, see *Equation (18)*. In terms of the determinant, we use the same approach considered in *Petkova et al., 2016*—in particular, concatenating $C^{\top}$ and $I$, the matrix $[C^{\top} \,|\, I]$ is orthogonal, so it can be shown that

$$\det(\Sigma) = \frac{\det(I^{\top} I) \det(C \Sigma C^{\top})}{\det(C C^{\top}) \det(I^{\top} \Sigma^{-1} I)}.$$

Rearranging terms and using the fact $\det(U^{-1}) = \det(U)^{-1}$ for any matrix $U$, we obtain:

$$\det(C \Sigma C^{\top})^{-1} = \frac{\det(I^{\top} I) \det(\Sigma^{-1})}{\det(C C^{\top}) \det(I^{\top} \Sigma^{-1} I)} = \frac{o}{I^{\top} \Sigma^{-1} I} \det(\Sigma^{-1}).$$

We have computed $\Sigma^{-1}$ in *Equation (14)*, so each of the terms above can be computed without any additional matrix multiplications. Finally, the signed graph incidence matrix $\Delta$ defined on the

edges of the graph is, by construction, highly sparse with $\mathcal{O}(d)$ many nonzero entries. Hence we implement sparse matrix multiplication to evaluate the penalty function $\phi_{\lambda,\alpha}(\boldsymbol{w})$ while avoiding the full-dimensional matrix-vector product.

## Estimating the edge weights under the exact likelihood model

When we developed the FEEMS model, we used the approximation $\frac{1}{2}f_j(k)(1-f_j(k)) \approx \sigma^2 \mu_j(1-\mu_j)$ for all SNPs $j$ and all nodes $k$ (see **Equation 4**) and estimated the residual variance $\sigma^2$ under the homogeneous isolation by distance model. The primary reason of using this approximation was primarily computational. While the approximation is not too strong if SNPs with rare allele frequencies are excluded, it is also critical that the estimation quality of the migration rates is not affected. In this subsection we introduce the inferring procedure of the migration rates under the exact likelihood model and compare it with FEEMS.

Note that without approximation, we can calculate the exact analytical form for the marginal likelihood of the estimated frequency as follows (after removing the SNP means):

$$\boldsymbol{C}\widehat{\boldsymbol{f}}_j \sim \sqrt{\mu_j(1-\mu_j)} \cdot \mathcal{N}_{o-1}\left(\boldsymbol{0}, \boldsymbol{C}\boldsymbol{A}\boldsymbol{L}^\dagger\boldsymbol{A}^\top\boldsymbol{C}^\top + \boldsymbol{C}\mathrm{diag}(\boldsymbol{n}^{-1})\boldsymbol{A}\mathrm{diag}\left(\left\{\frac{1-L^\dagger_{kk}}{2}\right\}^d_{k=1}\right)\boldsymbol{A}^\top\boldsymbol{C}^\top\right), \qquad (19)$$

where $\{a_k\}^d_{k=1}$ represents the vector $\boldsymbol{a}=(a_1,\ldots,a_d)$. Compared to the model **Equation (5)**, this expression does not introduce the unknown residual variance parameter $\sigma^2$ and instead each node has its own residual parameter given by $(1-L^\dagger_{kk})/2$. Because the residual parameters must be positive, this means that we have to search for the graphs that ensure $L^\dagger_{kk} \leq 1$ for all nodes $k$. With that said, we can consider the following constrained optimization problem:

$$\widehat{\boldsymbol{w}} = \underset{l \leq \boldsymbol{w} \leq \boldsymbol{u}}{\mathrm{argmin}}\left\{\ell_{exact}(\boldsymbol{w};\boldsymbol{C}\widehat{\boldsymbol{\Sigma}}\boldsymbol{C}^\top) + \phi_{\lambda,\alpha}(\boldsymbol{w}) : L^\dagger_{kk} \leq 1 \text{ for all } k \in \mathcal{V}\right\}, \qquad (20)$$

where $\ell_{exact}$ is the negative log-likelihood function based on **Equation (19)** and $\phi_{\lambda,\alpha}$ is the smooth penalty function defined earlier. The main difficulty of solving **Equation (20)** is that enforcing the constraint $L^\dagger_{kk} \leq 1$ for all nodes $k \in \mathcal{V}$, requires full computation of the pseudo-inverse of a $d \times d$ matrix $\boldsymbol{L}$ which is computationally demanding. We instead relax the constraint and consider the following form as a proxy for optimization **Equation (20)**:

$$\widehat{\boldsymbol{w}} = \underset{l \leq \boldsymbol{w} \leq \boldsymbol{u}}{\mathrm{argmin}}\left\{\ell_{exact}(\boldsymbol{w};\boldsymbol{C}\widehat{\boldsymbol{\Sigma}}\boldsymbol{C}^\top) + \phi_{\lambda,\alpha}(\boldsymbol{w}) : L^\dagger_{kk} \leq 1 \text{ for all observed nodes } k\right\}. \qquad (21)$$

Note that the constraint $L^\dagger_{kk} \leq 1$ is now placed at the observed nodes only, which can lead to computational savings if $o \ll d$. The problem **Equation (21)** can be solved efficiently using any gradient-based algorithms where we can calculate the gradient of $\ell_{exact}$ with respect to $\boldsymbol{L}$ as

$$\frac{\partial\ell_{exact}(\boldsymbol{w})}{\partial\mathrm{vec}(\boldsymbol{L})} = p \cdot \mathrm{vec}\left(\boldsymbol{L}^{-1}_{\mathrm{full}}\boldsymbol{A}^\top\boldsymbol{M}\boldsymbol{A}\boldsymbol{L}^{-1,\top}_{\mathrm{full}}\right) - p \cdot \mathrm{diag}(\boldsymbol{M})^\top\mathrm{diag}((2\boldsymbol{n})^{-1})\boldsymbol{N},$$

where $\boldsymbol{M}$ is a $o \times o$ matrix defined in **Equation (18)**, and $\boldsymbol{N}$ is a $o \times d^2$ matrix whose rows correspond to the observed subsets of the rows of the $d^2 \times d^2$ matrix $\boldsymbol{L}^{-1}_{\mathrm{full}} \otimes \boldsymbol{L}^{-1}_{\mathrm{full}}$.

**Appendix 1—figure 12** shows the result when the penalized maximum likelihood **Equation (21)** is applied to the North American wolf dataset with a setting of $\lambda = 2.06$ (the same value of $\lambda$ as given in **Figure 4**) and $\alpha = 1/\widehat{w}_0$, where $\widehat{w}_0$ is the solution for the 'constant-$w$' model. We can see that the resulting estimated migration surfaces are qualitatively similar to that shown in **Figure 4**. We also observed similar results between FEEMS and the penalized maximum likelihood **Equation (21)** across multiple datasets. On the other hand, we found that at the fitted surface the residual variances $1-L^\dagger_{kk}$ are not always positive because the constraints are enforced only at the observed nodes. This is problematic because it can cause the model to be ill-defined at the unobserved nodes and make the algorithm numerically unstable. Note that FEEMS avoids this issue by decoupling the residual variance parameter $\sigma^2$ from the graph-related parameters $\boldsymbol{w}$. The resulting model **Equation (6)**

also has more resemblance to spatial coalescent model used in EEMS (*Petkova et al., 2016*), and we thus recommend using FEEMS as a primary method for inferring migration rates.

## Jointly estimating the residual variance and edge weights

One simple strategy we have used throughout the paper was to fit $\sigma^2$ first under a model of homogeneous isolation by distance and prefix the estimated residual variance to the resulting $\hat{\sigma}^2$ for later fits of the effective migration rates. Alternatively, we can consider estimating the unknown residual variance simultaneously with the edge weights, instead of prefixing it from the estimation of the null model—the hope here is to simultaneously correct the model misspecification and allow for improving model fit to the data. To develop the framework for simultaneous estimation of the residual variance and edge weights, let us consider a model that generalizes both *Equation (6)* and *Equation (19)*, that is,

$$p \cdot \boldsymbol{C}\widehat{\boldsymbol{\Sigma}}\boldsymbol{C}^\top \sim \mathcal{W}_{o-1}\left(\boldsymbol{C}\boldsymbol{A}\boldsymbol{L}^\dagger\boldsymbol{A}^\top\boldsymbol{C}^\top + \boldsymbol{C}\operatorname{diag}(\boldsymbol{n}^{-1})\boldsymbol{A}\operatorname{diag}(\sigma^2)\boldsymbol{A}^\top\boldsymbol{C}^\top, p\right), \tag{22}$$

where $\sigma^2$ is a $d \times 1$ vector of node specific residual variance parameters, that is, each deme has its own residual parameter $\sigma_k$. If the parameters $\sigma_k$'s are assumed to be the same across nodes, this reduces to the FEEMS model *Equation (6)* while setting $\sigma_k = (1 - L_{kk}^\dagger)/2$ gives the model *Equation (19)*. Then we solve the following optimization problem

$$\widehat{\boldsymbol{w}}, \widehat{\sigma}^2 = \underset{\boldsymbol{l} \leq \boldsymbol{w} \leq \boldsymbol{u}, \sigma^2 > 0}{\arg\min} \ \ell_{joint}(\boldsymbol{w}, \sigma^2; \boldsymbol{C}\widehat{\boldsymbol{\Sigma}}\boldsymbol{C}^\top) + \phi_{\lambda,\alpha}(\boldsymbol{w}),$$

where $\ell_{joint}$ is the negative log-likelihood function based on *Equation (22)*. Note that the residual variances and edge weights are both searched in the optimization for finding the optimal solutions. To solve the problem, we can use the quasi-newton algorithm for optimizing the objective function.

*Appendix 1—figure 13* shows the fitted graphs with different strategies of estimating the residual variances. *Appendix 1—figure 13A* shows the result when the model has a single residual variance $\sigma^2$, and *Appendix 1—figure 13B* shows the result when the residual variances are allowed to vary across nodes. In both cases, estimating the residual variances jointly with the edge weights yields similar and comparable outputs to the default setting of prefixing it from the null model (*Figure 4*), except that we can further observe reduced effective migration around Queen Elizabeth Islands as shown in *Appendix 1—figure 13B*. In EEMS, in order to estimate the genetic diversity parameters for every spatial location, which play a similar role as the residual variances in FEEMS, a Voronoi-tessellation prior is placed to encourage sharing of information across adjacent nodes and prevent over-fitting. Similarly, we can place the spatial smooth penalty on the residual variances (i.e. $\phi_{\lambda,\alpha}$ defined on the variable $\sigma^2$), but it introduces additional hyperparameters to tune, without substantially improving the model's fit to the data. In this work, we choose to fit the single residual variance $\sigma^2$ under the null model and prefix it as a simple but effective strategy with apparent good empirical performance.

## Edge versus node parameterization

One of the novel features of FEEMS is its ability to directly fit the edge weights of the graph that best suit the data. This direct edge parameterization may increase the risk of model's overfitting, but also allows for more flexible estimation of migration histories. Furthermore, as seen in *Figure 2* and *Appendix 1—figure 2*, it has potential to recover anisotropic migration processes. This is in contrast to EEMS wherein every spatial node is assigned an effective migration parameter $m_k$ and a migration rate on each edge joining nodes $k$ and $\ell$ is given by the average effective migration $w_{k\ell} = (m_k + m_\ell)/2$. Not surprisingly, by assigning each edge to be the average of connected nodes, a form of implicit spatial regularization is imposed because multiple edges connected to the same node would average that node's parameter value. In some cases, this has the desirable property of imposing an additional degree of similarity across edge weights, but at the same time it also restricts the model's capacity to capture a richer set of structure present in the data (e.g. *Petkova et al., 2016*, Supplementary Figure 2). To be concrete, *Appendix 1—figure 15* displays two different fits of FEEMS based on edge parameterization (*Appendix 1—figure 15A*) and node parameterization

(*Appendix 1—figure 15B*), run on a previously published dataset of human genetic variation from Africa (see *Peter et al., 2020* for details on the description of the dataset). Running FEEMS with a node-based parameterization is straightforward in our framework—all we have to do is to reparameterize the edge weights by the average effective migration and solve the corresponding optimization problem (Optimization) with respect to $m$. It is evident from the results that FEEMS with edge parameterization exhibits subtle correlations that exist between the annotated demes in the figure, whereas node parameterization fails to recover them. We also compare the model fit of FEEMS to the observed genetic distance (*Appendix 1—figure 16*) and find that edge-based parameterization provides a better fit to the African dataset. *Appendix 1—figure 17* further demonstrates that in the coalescent simulations with anisotropic migration, the node parameterization is unable to recover the ground truth of the underlying migration rates even when the nodes are fully observed.

## Smooth penalty with $\ell_1$ norm

FEEMS's primary optimization objective (see *Equation 9*) is:

$$\underset{l \leq w \leq u}{\text{Minimize}} \, \ell(w, \sigma^2; C\widehat{\Sigma}C^\top) + \phi_{\lambda,\alpha}(w),$$

where the spatial smoothness penalty is given by an $\ell_2$-based penalty function: $\phi_{\lambda,\alpha}(w) = \frac{\lambda}{2}\|\Delta \log(e^{\alpha w} - I)\|_2^2$. It is well known that an $\ell_1$-based penalty can lead to a better local adaptive fitting and structural recovery than $\ell_2$-based penaltyies (*Wang et al., 2016*), but at the cost of handling non-smooth objective functions that are often computationally more challenging. In a spatial genetic dataset, one major challenge is to deal with the relatively sparse sampling design where there are many unobserved nodes on the graph. In this statistically challenging scenario, we found that an $\ell_2$-based penalty allows for more accurate and reliable estimation of the geographic features.

Specifically, writing $\phi_{\lambda,\alpha}^{\ell_1}(w) = \lambda\|\Delta \log(e^{\alpha w} - I)\|_1$, we considered the alternate following composite objective function:

$$\ell(w, \sigma^2; C\widehat{\Sigma}C^\top) + \phi_{\lambda,\alpha}^{\ell_1}(w). \tag{23}$$

To solve *Equation (23)*, we apply linearized alternating direction method of multipliers (ADMM) (*Boyd, 2010*), a variant of the standard ADMM algorithm, that iteratively optimizes the augmented Lagrangian over the primal and dual variables. The derivation of the algorithm is a standard calculation so we omit the detailed description of the algorithm. As opposed to the common belief about the effectiveness of the $\ell_1$ norm for structural recovery, the recovered graph of FEEMS using $\ell_1$-based smooth penalty shows less accurate reconstruction of the migration patterns, especially when the sampling design has many locations with missing data on the graph (*Appendix 1—figure 18A*, *Appendix 1—figure 19H*). We can see that the $\ell_1$-based penalty function is not able to accurately estimate edge weights at regions with little data, partially due to its local adaptation, in contrast to the $\ell_2$-based method that considers regularization more globally. This suggests that in order to use the $\ell_1$ penalty $\phi_{\lambda,\alpha}^{\ell_1}(w)$ in the presence of many missing nodes, one may need an additional degree of regularization that encourages global smoothness of the graph's edge weights, such as a combination of $\phi_{\lambda,\alpha}^{\ell_1}(w)$ and $\phi_{\lambda,\alpha}(w)$ (in the same spirit as elastic net [*Zou and Hastie, 2005*]), or $\phi_{\lambda,\alpha}^{\ell_1}(w)$ on top of node-based parameterization (see *Appendix 1—figure 18B*).

## Coalescent simulations with weak migration

In *Figure 2*, we evaluated FEEMS by applying it to 'out-of-model' coalescent simulations. In these simulations, we generated genotype data under a coalescent model with structured meta-populations organized on a spatial triangular lattice. In a relatively 'strong' heterogeneous migration scenario (*Figure 2D,E,F*), we set the coalescent migration rate to be an order of magnitude lower (10-fold) in the center of the spatial grid than on the left and right regions, emulating a depression in gene-flow caused, for example, by a mountain range or body of water. The variation in migration rates should create a spatially varying covariance structure in the genetic variation data. To get a sense of the level of genetic divergence implied by this simulation setting, we visualized Wright's

fixation index ($F_{ST}$, Patterson's estimator [*Patterson et al., 2012*]) plotted against the geographic distance between nodes (*Appendix 1—figure 20*). We see in the strong heterogeneous migration simulation there is a clear signal of two clusters of data points (*Appendix 1—figure 20B*). These clusters correspond to pairwise $F_{ST}$ comparisons of two nodes on the same side of the central depression in gene flow, where gene-flow roughly follows a homogeneous 'isolation-by-distance' like pattern, or two nodes across the central depression where gene-flow is reduced, hence increasing the expected $F_{ST}$ between such nodes.

While simulating this strong reduction of gene-flow provides an illustrative and clear example where FEEMS has a lot of signal for accurate inference, we wanted to understand the qualitative performance of FEEMS in an less idealized scenario with weaker signal. To this end, we performed coalescent simulations with only a 25% reduction of gene-flow in the center of the habitat (*Appendix 1—figure 21*). In *Appendix 1—figure 21A*, when all the nodes are observed on the spatial graph, FEEMS is still able to detect this subtle reduction of gene-flow. While FEEMS is able to detect this signal, there remain particularly erroneous estimates among the lower than average edge weights, implying the fit could benefit from additional smoothing by increasing the level regularization on the smoothness penalty. In contrast to the strong heterogeneous migration simulations, we see that the pairwise $F_{ST}$ in this weak migration scenario does not obviously show a 'clustering' like effect in the data (*Appendix 1—figure 20A*). The average $F_{ST}$ between all pairs of demes is approximately three times lower (mean $F_{ST}$ = .1175 for the weak heterogeneity simulation versus mean $F_{ST}$ = 0.3411 for the strong heterogeneity simulation). When the nodes are sparsely observed on the graph in this weak migration simulation, we see that the FEEMS output is overly smooth (*Appendix 1—figure 21B*). In the absence of data and thus a weak signal for spatial variation in migration, a smooth visualization is arguably a sensible outcome given the regularization acts like a prior distribution favoring spatial homogeneity in levels of effective migration.

In practice, weak population structure can be more accurately dissected when increasing the number of informative SNPs included in the analysis (*Novembre and Peter, 2016*). In conjunction with running FEEMS, we recommend for users to create exploratory visualizations such as variograms and PCA bi-plots to assess the level of population structure in their data, and to consider the number of SNPs used in the analysis.

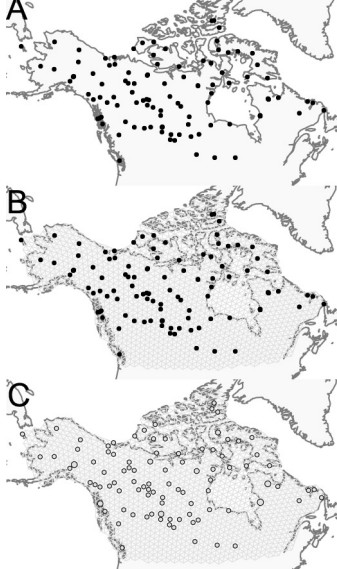

**Appendix 1—figure 1.** Visualization of grid construction and node assignment: (**A**) Map of sample coordinates (black points) from a dataset of gray wolves from North America. The input to FEEMS are latitude and longitude coordinates as well as genotype data for each sample. (**B**) Map of sample coordinates with an example dense spatial grid. The nodes of the grid represent sub-populations and the edges represent local gene-flow between adjacent sub-populations. (**C**) Individuals are

*Appendix 1—figure 1 continued on next page*

*Appendix 1—figure 1 continued*

assigned to nearby nodes (sub-populations) and summary statistics (e.g. allele frequencies) are computed for each observed location.

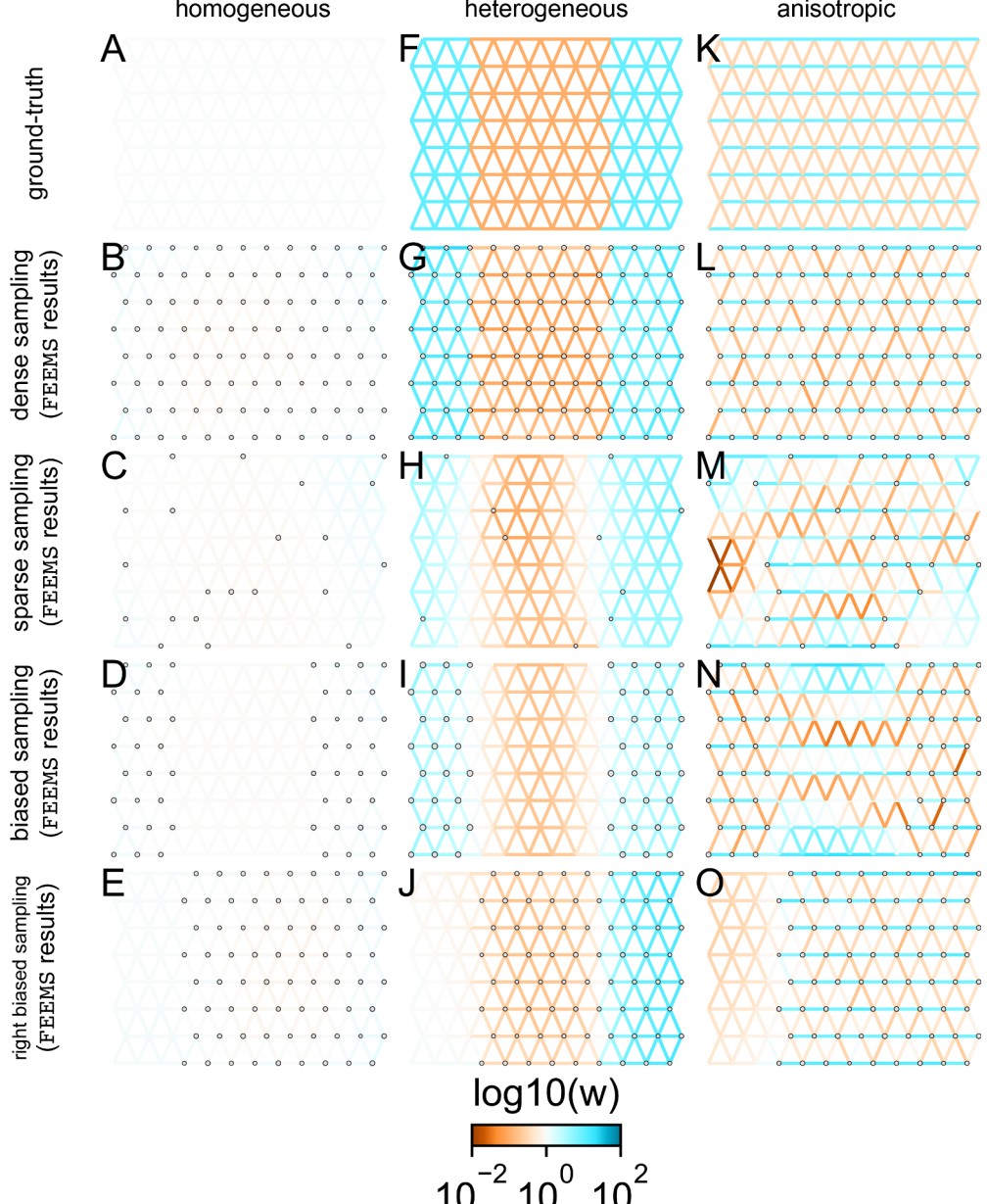

**Appendix 1—figure 2.** Application of FEEMS to an extended set of coalescent simulations: We display an extended set of coalescent simulations with multiple migration scenarios and sampling designs. The sample sizes across the grid are represented by the size of the gray dots at each node. The migration rates are obtained by solving FEEMS objective function *Equation 9* where the the smoothness parameter λ was selected using leave-one-out cross-validation. (**A, F, K**) display the ground truth of the underlying migration rates. (**B, G, L**) shows simulations where there is no missing data on the graph. (**C, H, M**) shows simulations with sparse observations and nodes missing at random. (**D, I, N**) shows simulations of biased sampling where there are no samples from the center of the simulated habitat. (**E, J, O**) shows simulations of biased sampling where there are only samples on the right side of the habitat.

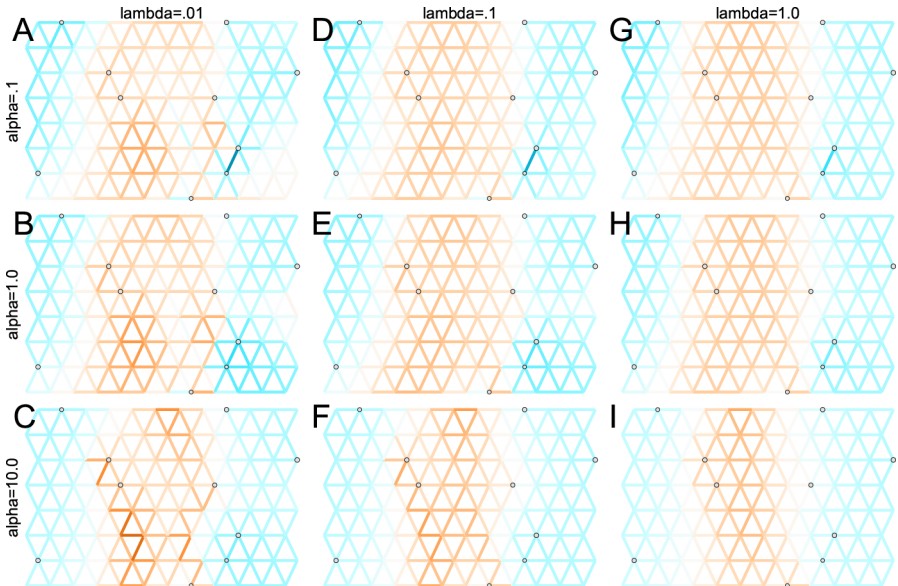

**Appendix 1—figure 3.** Application of FEEMS to a heterogeneous migration scenario with a 'missing at random' sampling design: We run FEEMS on coalescent simulation with a non-homogeneous process while varying hyperparameters λ (rows) and α (columns). We randomly sample individuals for 20% of nodes. When λ grows, the fitted graph becomes overall smoother, whereas α effectively controls the degree of similarity among low migration rates.

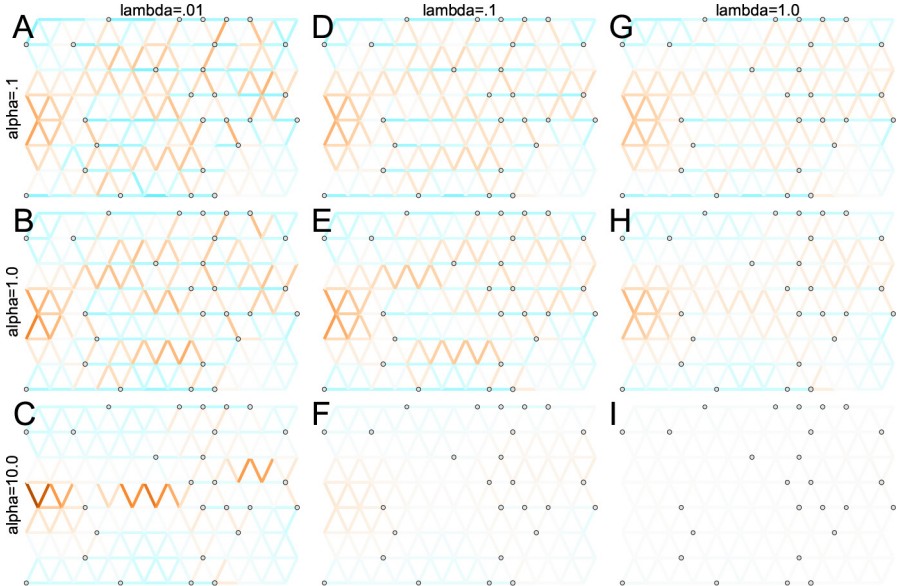

**Appendix 1—figure 4.** Application of FEEMS to an anisotropic migration scenario with a 'missing at random' sampling design: We run FEEMS on coalescent simulation with an anisotropic process while varying hyperparameters λ (rows) and α (columns). We randomly sample individuals for 20% of nodes. When λ grows, the fitted graph becomes overall smoother, whereas α effectively controls the degree of similarity among low migration rates.

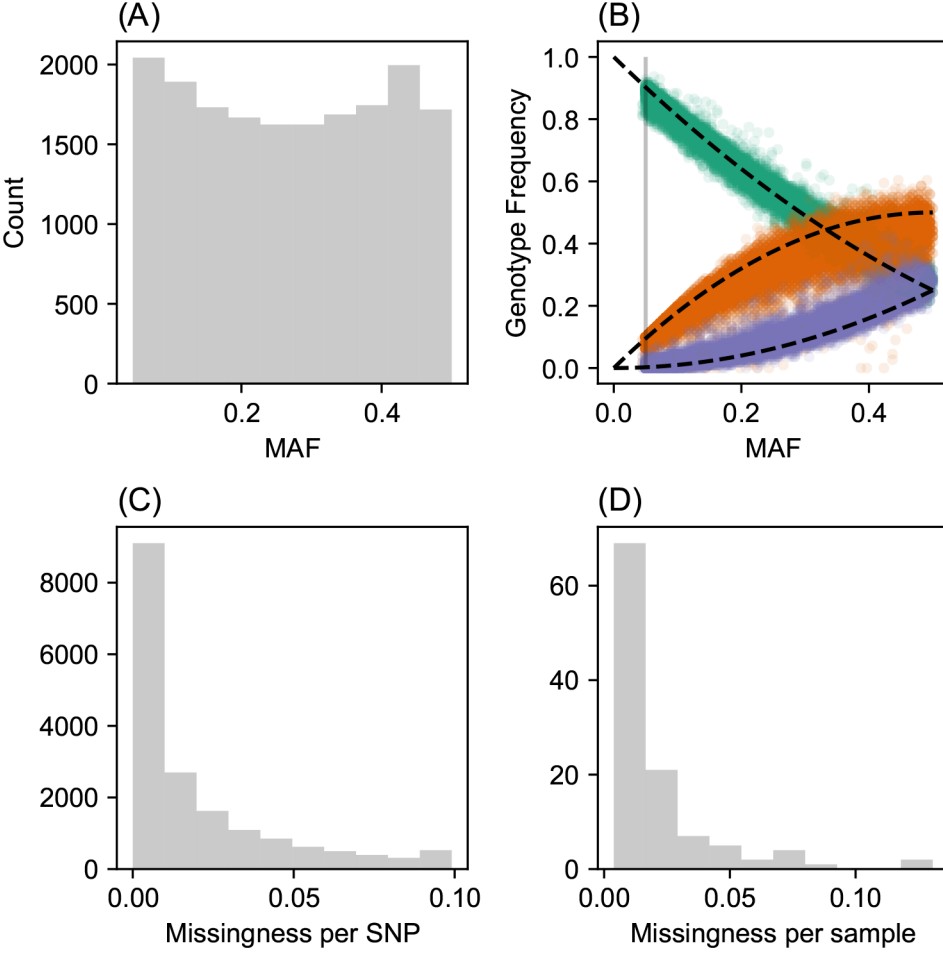

**Appendix 1—figure 5.** SNP and individual quality control. (**A**) displays a visualization of the sample site frequency spectrum. Specifically, we display a histogram of minor allele frequencies across all SNPs. We see a relatively uniform histogram which reflects the ascertainment of common SNPs on the array that was designed to genotype gray wolf samples. (**B**) visualization of allele frequencies plotted against genotype frequencies. Each point represents a different SNP and the colors represent the three possible genotype values. The black dashed lines display the expectation as predicted from a simple binomial sampling model i.e., Hardy-Weinberg equilibrium. (**C**) displays a histogram of the missingness fraction per SNP. We observe the missingness tends to be relatively low for each SNP. (**D**) displays a histogram of the missingness fraction per sample. Generally, the missingness tends to be low for each sample.

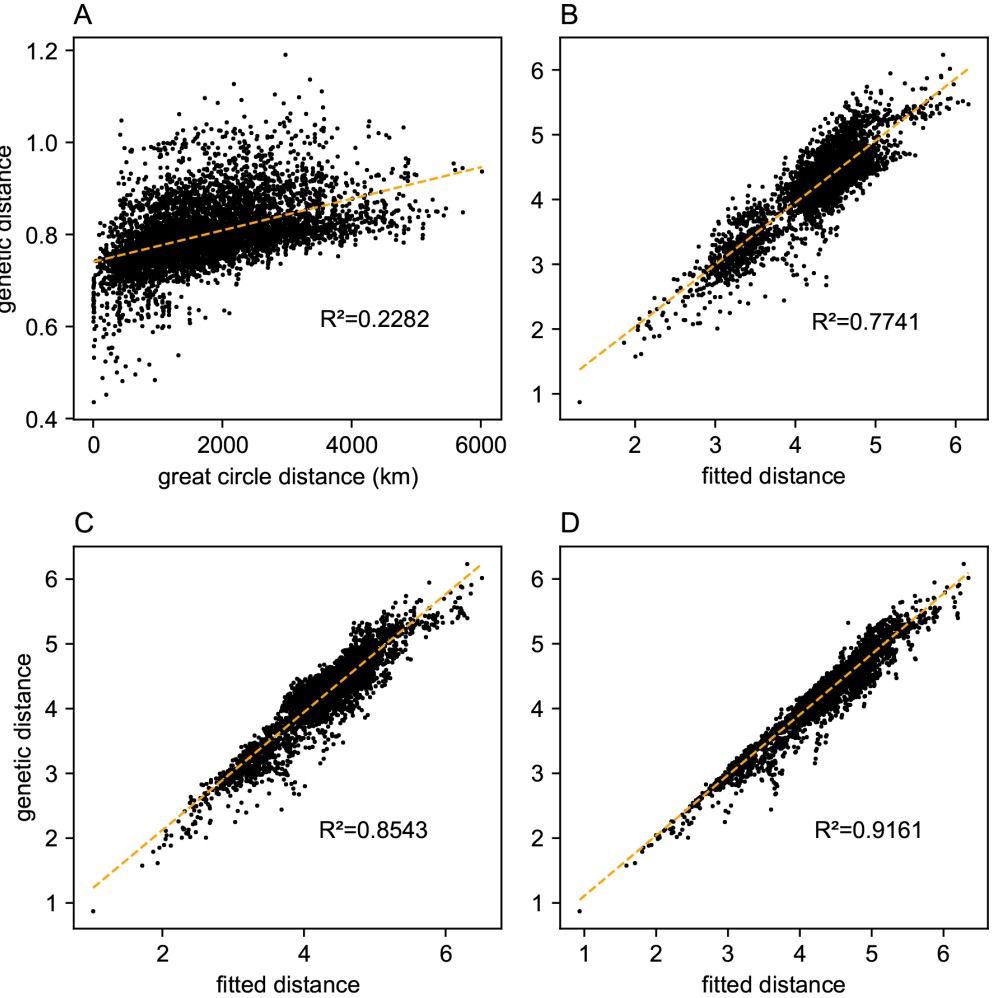

**Appendix 1—figure 6.** Comparing predictions of observed genetic distances: We display different predictions of observed genetic distances using geographic distance or the fitted genetic distance output by FEEMS. (**A**) The x-axis displays the geographic distance between two individuals, as measured by the great circle distance (haversine distance). The y-axis displays the squared Euclidean distance between two individuals averaged over all SNPs. (**B–D**) The x-axis displays the fitted genetic distance as predicted by the FEEMS model and y-axis displays the squared Euclidean distance between two sub-populations averaged over all SNPs. For (**B–D**), we display the fit of $\lambda$ getting subsequently smaller, $\lambda = 100, 2.06, 0.04$ (the same values of $\lambda$ used in *Figure 3A,B,C*), and as expected the fit appears better because we tolerate more complex surfaces and we are not evaluating the fit on out-of-sample data.

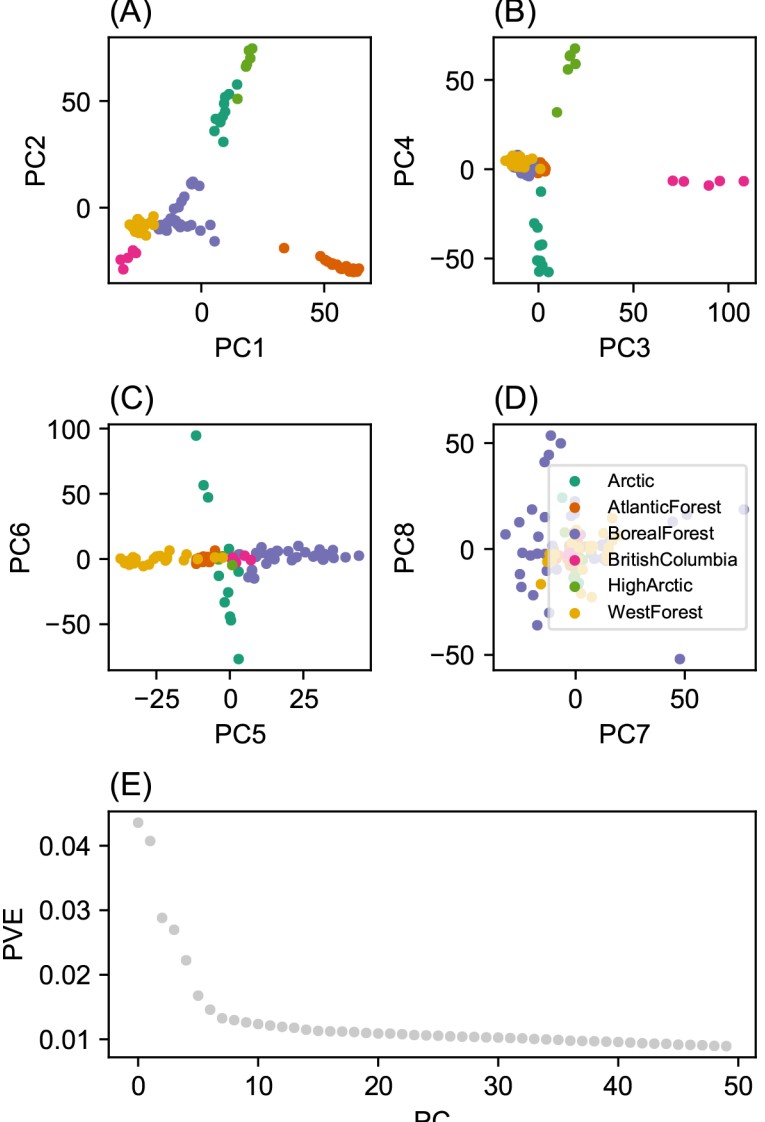

**Appendix 1—figure 7.** Summary of top axes of genotypic variation: We display a visual summary of Principal Components Analysis (PCA) applied to the normalized genotype matrix from the North American gray wolf dataset. (A–D) displays PC bi-plots of the top seven PCs plotted against each other. The colors represent predefined ecotypes defined in *Schweizer et al., 2016*. We can see that the top PCs delineate these predefined ecotypes. (E) shows a 'scree' plot with the proportion of variance explained for each of the top 50 PCs. As expected by genetic data (*Patterson et al., 2006*), the eigenvalues of the genotype matrix tend to be spread over many PCs.

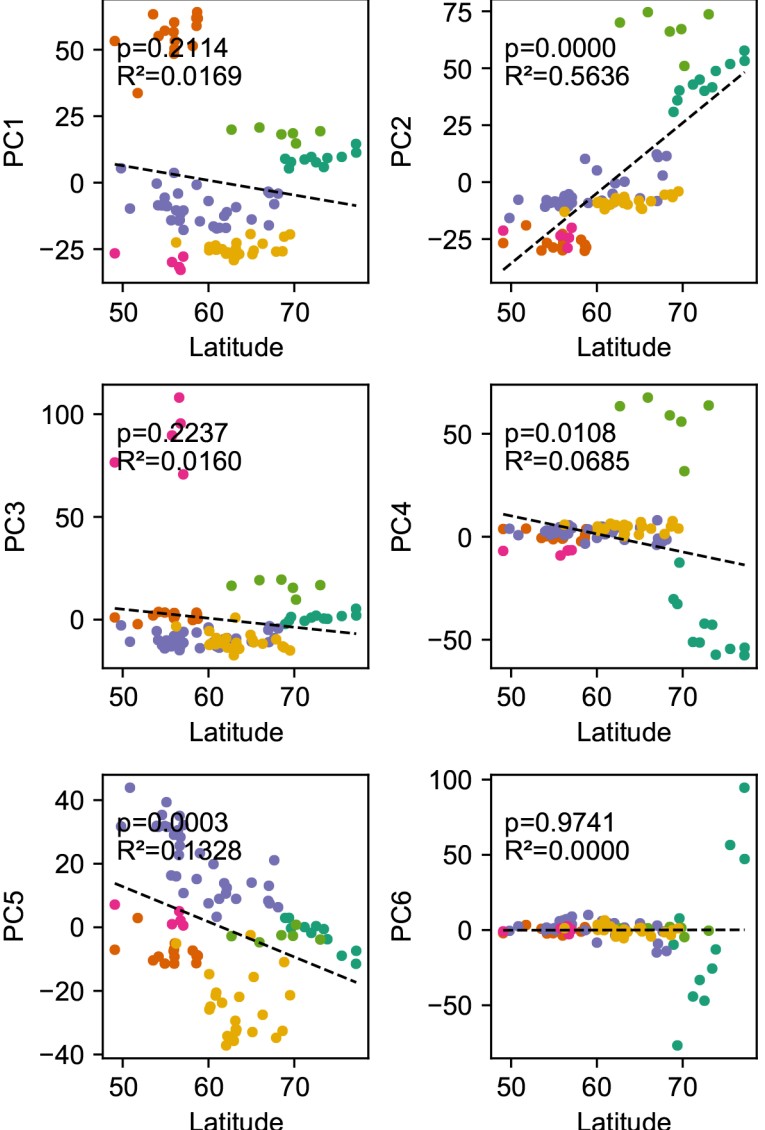

**Appendix 1—figure 8.** Relationship between top axes of genetic variation and latitude: In each sub-panel, we plot the PC value against latitude for each sample in gray the wolf dataset. We see many of the top PCs are significantly correlated with latitude as tested by linear regression.

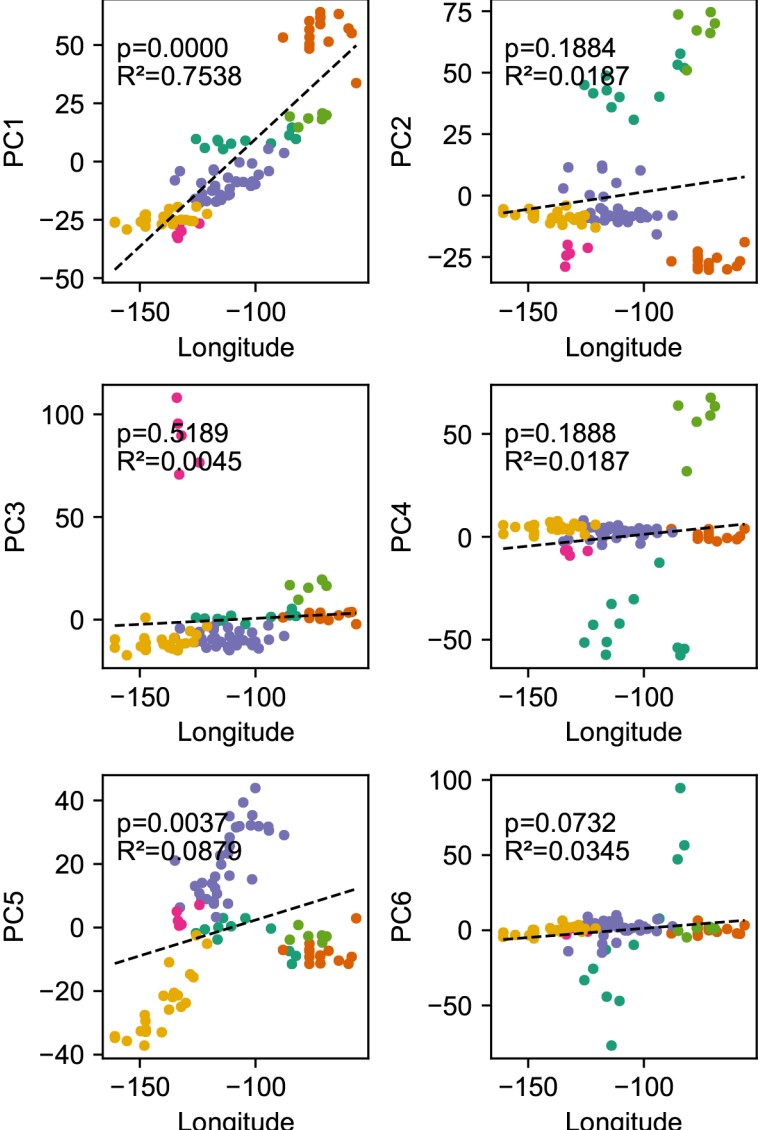

**Appendix 1—figure 9.** Relationship between top axes of genetic variation and longitude: In each sub-panel, we plot the PC value against longitude for each sample in the gray wolf dataset. We see many of the top PCs are significantly correlated with longitude as tested by linear regression.

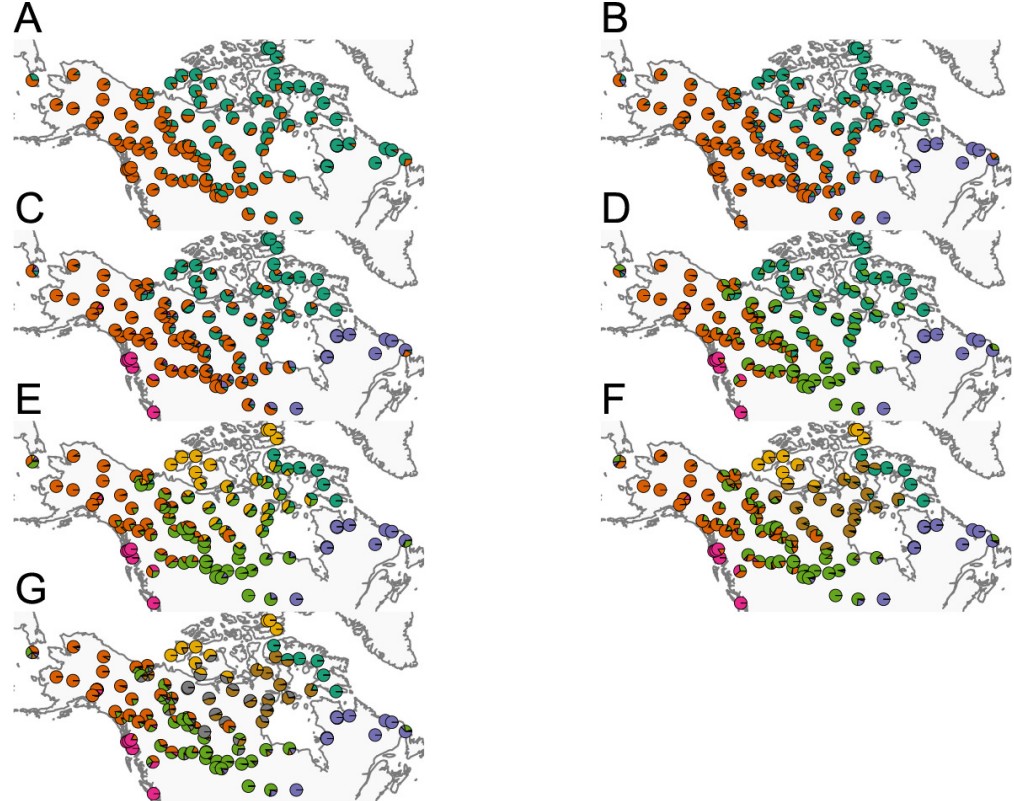

**Appendix 1—figure 10.** Summary of ADMIXTURE results: (**A–G**) Visualization of ADMIXTURE results for $K = 2$ to $K = 8$. We display admixture fractions for each sample as colored slices of the pie chart on the map. For each $K$, we ran five replicate runs of ADMIXTURE and in this visualization, we display the solution that achieves the highest likelihood amongst the replicates. The ADMIXTURE results qualitatively reveal a spatial signal in the data as admixture fractions tend to be spatially clustered.

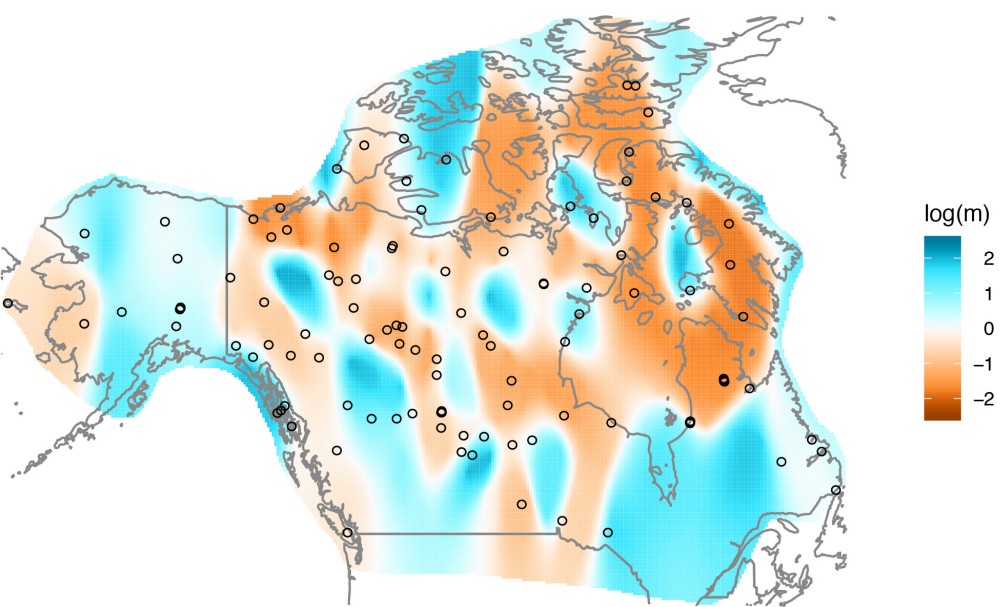

**Appendix 1—figure 11.** Application of EEMS to the North American gray wolf dataset: We display a visualization of EEMS applied to the North American gray wolf dataset. The more orange colors represent lower than average effective migration on the log-scale and the more blue colors represent higher than average effective migration on the log-scale. The results of EEMS are qualitatively similar to FEEMS when lower regularization penalties are applied.

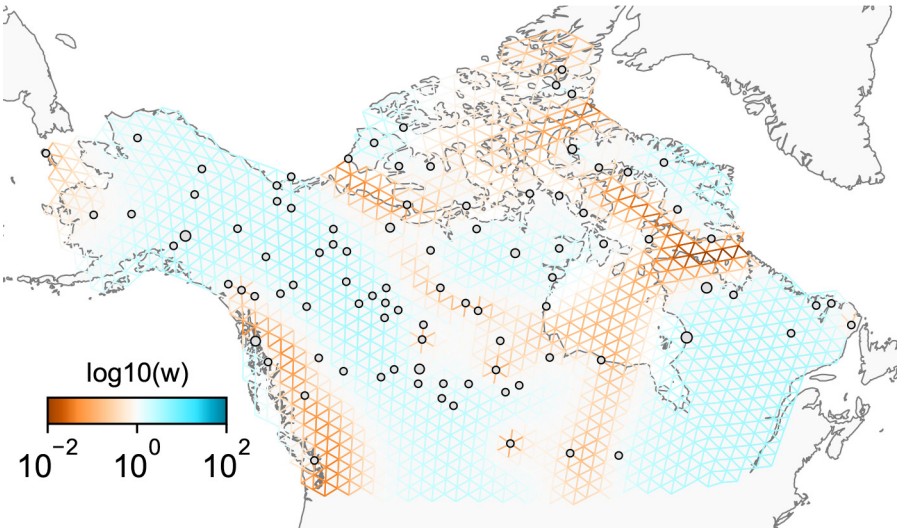

**Appendix 1—figure 12.** Application of FEEMS on the North American gray wolf dataset with an exact likelihood model: We display the fit of FEEMS based on the formulation *Equation (21)* to the North American gray wolf dataset. This fit corresponds to a setting of tuning parameters at $\lambda = 2.06$ and $\alpha = 1/\widehat{w}_0$. Additionally, we set the lower bound of the edge weights to $l = 10^{-6}$, to ensure that the diagonal elements of $L$ does not become too small—this has an implicit effect on $L_{kk}^{\dagger}$, preventing it from blowing up at unobserved nodes. The more orange colors represent lower than average effective migration on the log-scale and the more blue colors represent higher than average effective migration on the log-scale. Visually, the result is comparable to that of the FEEMS fit (*Figure 4*) based on the approximate formulation (Optimization).

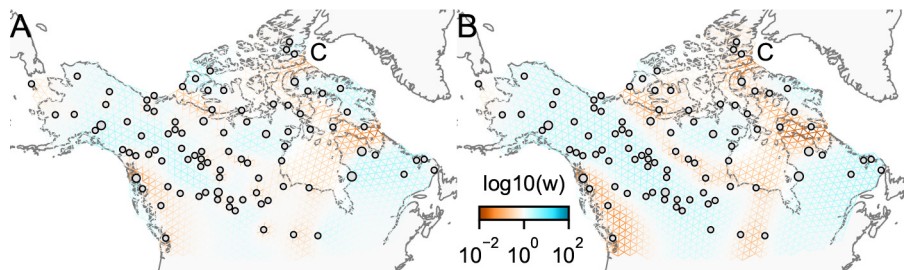

**Appendix 1—figure 13.** Application of FEEMS on the North American gray wolf dataset with joint estimation of the residual variances and graph's edge weights: We show visualizations of fits of FEEMS to the North American gray wolf dataset when the residual variances and edge weights of the graph are jointly estimated. Both fits correspond to a setting of tuning parameters at $\lambda = 2.06$ and $\alpha = 1/\widehat{w}_0$. (**A**) displays the estimated effective migration surfaces where every deme shares a single residual parameter σ. (**B**) displays the estimated effective migration surfaces where each node has its own residual parameter $\sigma_k$. Both approaches yield similar results to the procedure that prefixes $\sigma$ from the homogeneous isolation by distance model (*Figure 4*). The node-specific residual parameters may allow for more flexible graphs to be fitted, and we can further observe reduced effective migration around C (Queen Elizabeth Islands) in (**B**).

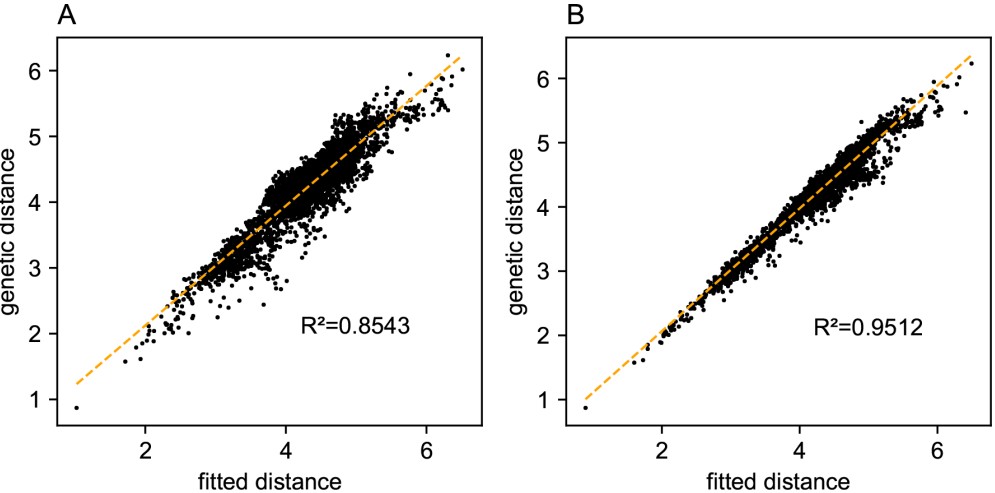

**Appendix 1—figure 14.** Relationship between fitted versus observed genetic dissimilarities on the North American gray wolf dataset: We display scatter plots of fitted genetic distance versus observed genetic distance from FEEMS fits on the gray wolf dataset. (**A**) Corresponds to the result shown in *Figure 4*. (**B**) Corresponds to the result shown in *Appendix 1—figure 13B*. The x-axis displays the fitted genetic distance as predicted by the FEEMS model and y-axis displays the squared Euclidean distance between two sub-populations averaged over all SNPs. The simple linear regression fit is shown in orange dashed lines and $R^2$ is given.

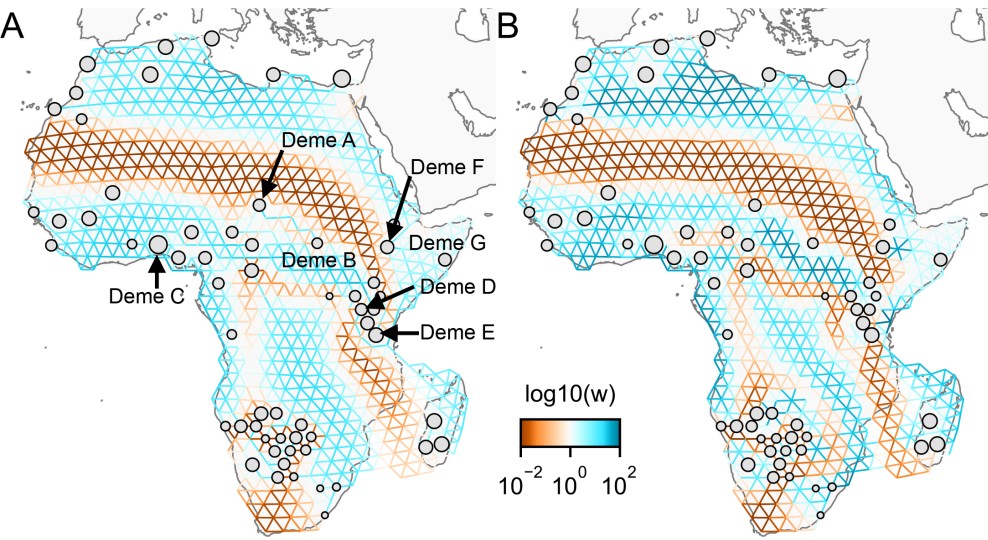

**Appendix 1—figure 15.** Application of FEEMS to a dataset of human genetic variation from Africa with different parameterization: We display visualizations of FEEMS to a dataset of human genetic variation from Africa with different parameterization of the graph's edge weights. See *Peter et al., 2020* for the description of the dataset. (**A**) displays the recovered graph under the edge parameterization. (**B**) displays the recovered graph under the node parameterization. Both parameterization have their own regularization parameters λ and α, but these parameters are not on the same scale. We set $\lambda = 0.2$ and $\alpha = 0.02$ for the node parameterization which is seen to yield similar results to those in *Peter et al., 2020*. For the edge parameterization, we set $\lambda = 0.5$ and $\alpha = 0.05$ so that the resulting graph reveals similar geographic structure to the node parameterization. We also set the lower bound $l = 10^{-6}$. From the plots, it is worth noting two important distinctions: (1) We see the migration surfaces shown in (**B**) recover sharper edge features while the migration surfaces in (**A**) are overall smoother. This is attributed to the fact that node parameterization has its own additional regularization effect on the edge weights, and in order to achieve similar degree of regularization strength for the edge parameterization, it needs a higher regularization parameters, which results in more blurring edges than the node parameterization. (2) When measuring correlation of the estimated allele frequencies among nodes, we find that Deme B is the node with the second highest correlation to Deme A, whereas Deme C (and nearby demes) is not as much correlated to Deme A compared to Deme B. Panel (**A**) reflects this feature by exhibiting a corridor between Deme A and Deme B and reduced gene-flow beneath that corridor. This reduced gene-flow disappears in (**B**), even if the regularization parameters are varied over a range of values. Additionally, Deme D is most highly correlated to Deme E, F, and G, and this is implicated by a long-range corridor connecting those demes appearing in Panel (**A**) while not shown in (**B**). These results suggest that the form of the node parameterization is perhaps too strong and in this case limits the model's ability to capture desirable geographic features that are subtle to detect.

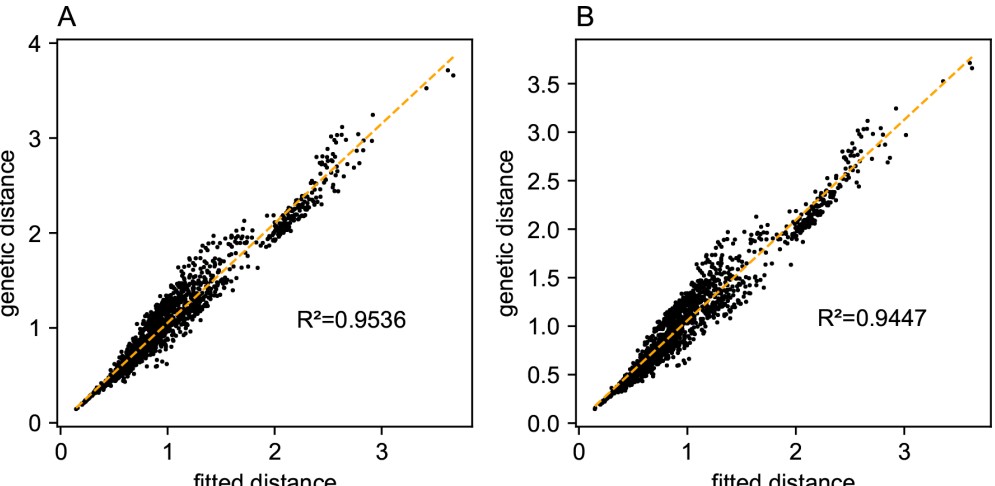

**Appendix 1—figure 16.** Relationship between fitted versus observed genetic dissimilarities on a dataset of human genetic variation from Africa: We display scatter plots of fitted genetic distance versus observed genetic distance from FEEMS fits on the African dataset. (**A**) Corresponds to the result shown in *Appendix 1—figure 15A*. (**B**) Corresponds to the result shown in *Appendix 1—figure 15B*. The x-axis displays the fitted genetic distance as predicted by the FEEMS model and y-axis displays the squared Euclidean distance between two sub-populations averaged over all SNPs. The simple linear regression fit is shown in orange dashed lines and $R^2$ is given.

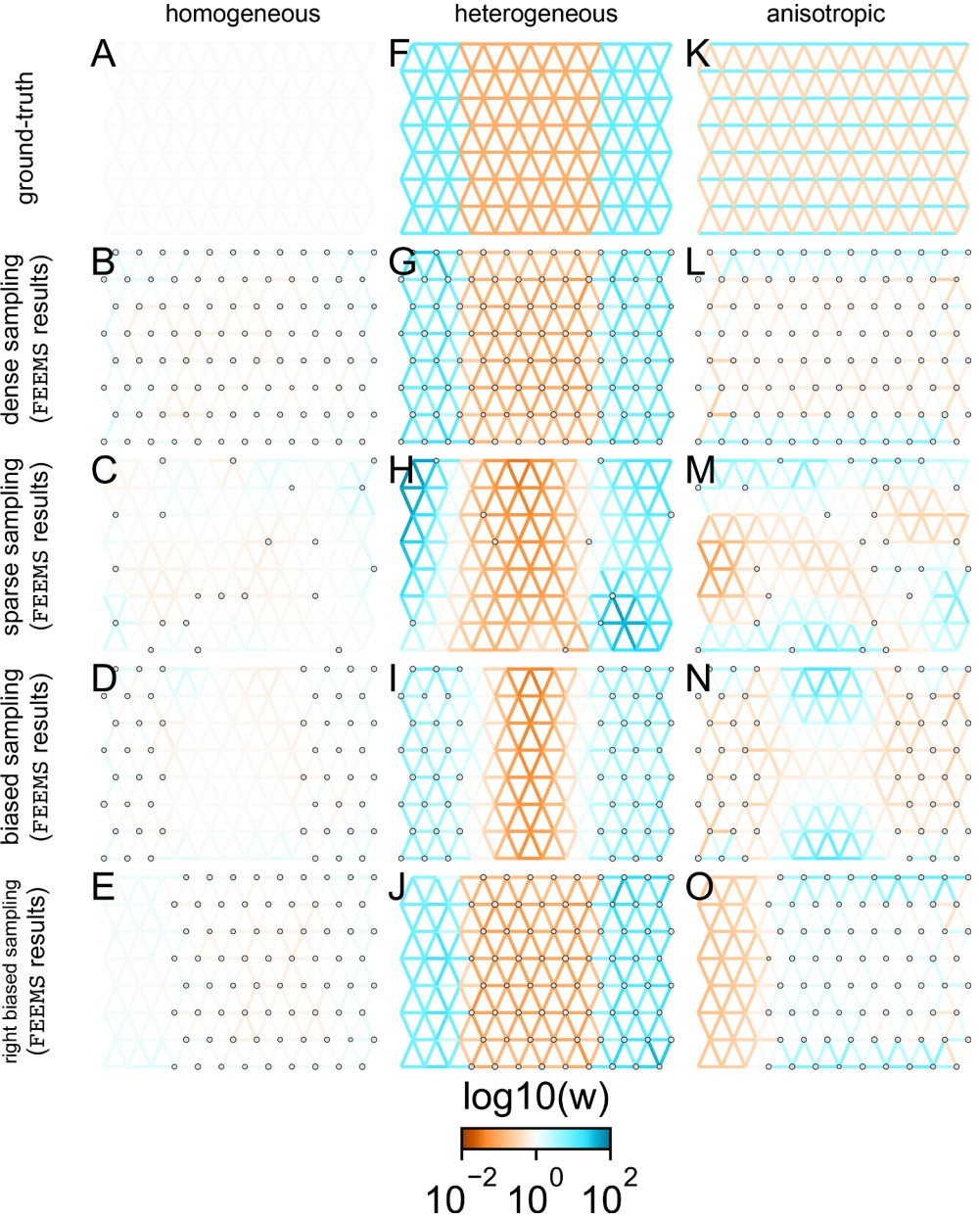

**Appendix 1—figure 17.** Application of FEEMS based on node parameterization to an extended set of coalescent simulations: We display an extended set of coalescent simulations with the same migration scenarios and sampling designs as *Appendix 1—figure 2*. The sample sizes across the grid are represented by the size of the gray dots at each node. The migration rates are obtained by solving the FEEMS objective function (Optimization) with node parameterization where the regularization parameters are specified at $\lambda = 5$ and $\alpha = 0.01$. (A, F, K) display the ground truth of the underlying migration rates. (B, G, L) shows simulations where there is no missing data on the graph. (C, H, M) shows simulations with sparse observations and nodes missing at random. (D, I, N) shows simulations of biased sampling where there are no samples from the center of the simulated habitat. (E, J, O) shows simulations of biased sampling where there are only samples on the right side of the habitat.

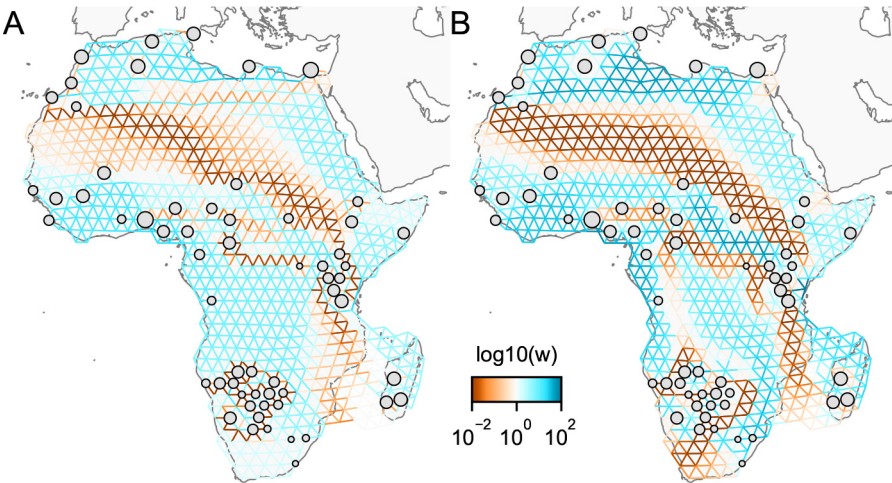

**Appendix 1—figure 18.** Application of $\ell_1$-norm-based FEEMS to a dataset of human genetic variation from Africa: We display visualizations of FEEMS to a dataset of human genetic variation from Africa with the $\ell_1$-based penalty function. See *Peter et al., 2020* for the description of the dataset. (**A**) displays the recovered graph under the edge parameterization with $\ell_1$ norm based penalty where the regularization parameters are specified at $\lambda = 0.05$ and $\alpha = 5$. (**B**) displays the recovered graph under the node parameterization with $\ell_1$ norm-based penalty where the regularization parameters are specified at $\lambda = 0.05$ and $\alpha = 1$. To minimize the objective *Equation (23)*, linearized ADMM is applied with 20,000 number of iterations. The lower bound is set to be $l = 10^{-6}$ for both parameterizations. Note that due to the high degrees of missingness, the estimated effective migration surfaces using solely $\ell_1$-based penalty exhibit many likely artifacts (e.g. high migration edges forming long paths, seen in panel A) unless an additional regularization is added to encourage global smoothness of the edge weights, such as a combination of $\ell_1$ norm penalty function and node parameterization as shown in (**B**).

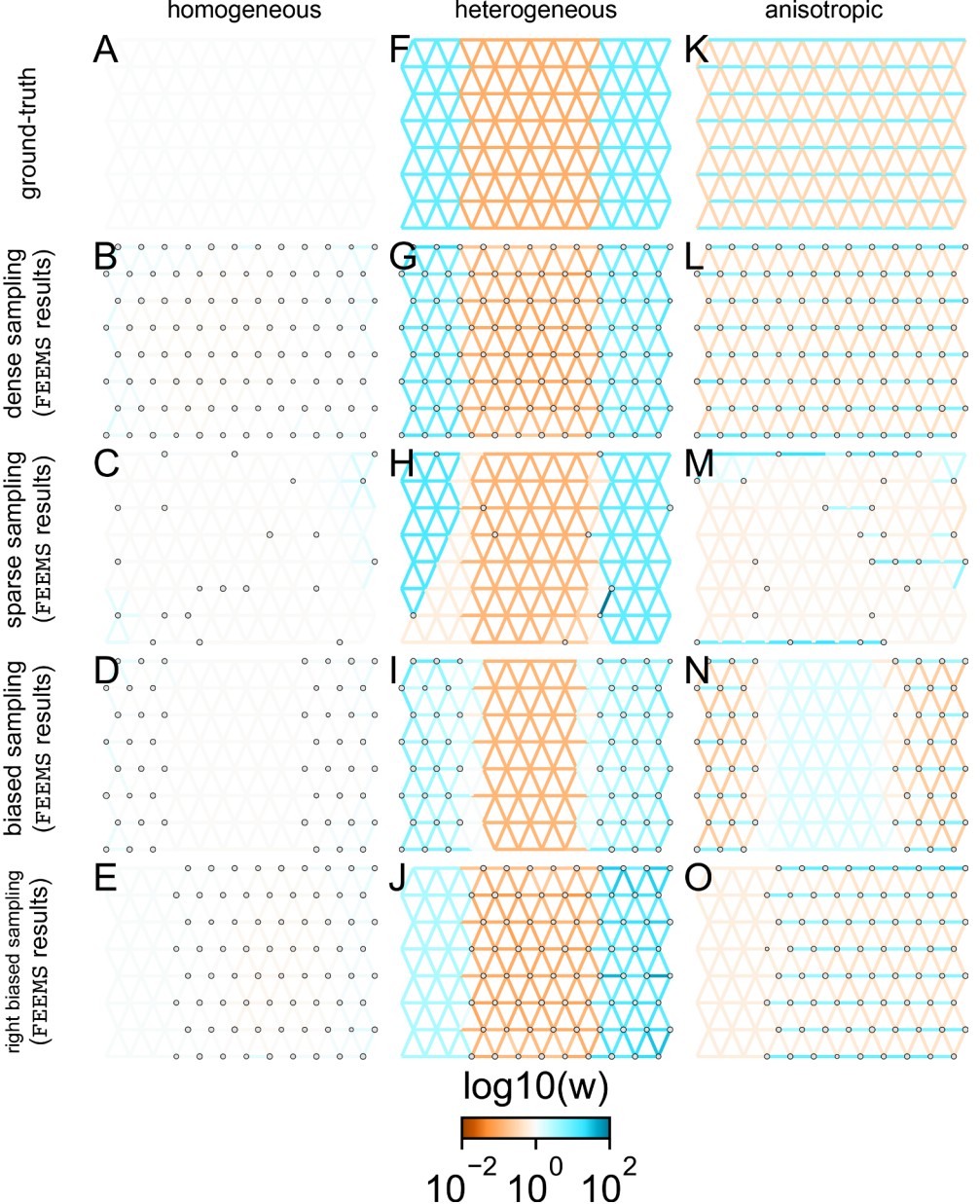

**Appendix 1—figure 19.** Application of $\ell_1$-norm-based FEEMS to an extended set of coalescent simulations: We display an extended set of coalescent simulations with the same migration scenarios and sampling designs as *Appendix 1—figure 2*. The sample sizes across the grid are represented by the size of the gray dots at each node. The migration rates are obtained by solving $\ell_1$ norm based FEEMS objective *Equation (23)* where the regularization parameters are specified at $\lambda = 5$ and $\alpha = 0.01$. (**A, F, K**) display the ground truth of the underlying migration rates. (**B, G, L**) shows simulations where there is no missing data on the graph. (**C, H, M**) shows simulations with sparse observations and nodes missing at random. (**D, I, N**) shows simulations of biased sampling where there are no samples from the center of the simulated habitat. (**E, J, O**) shows simulations of biased sampling where there are only samples on the right side of the habitat.

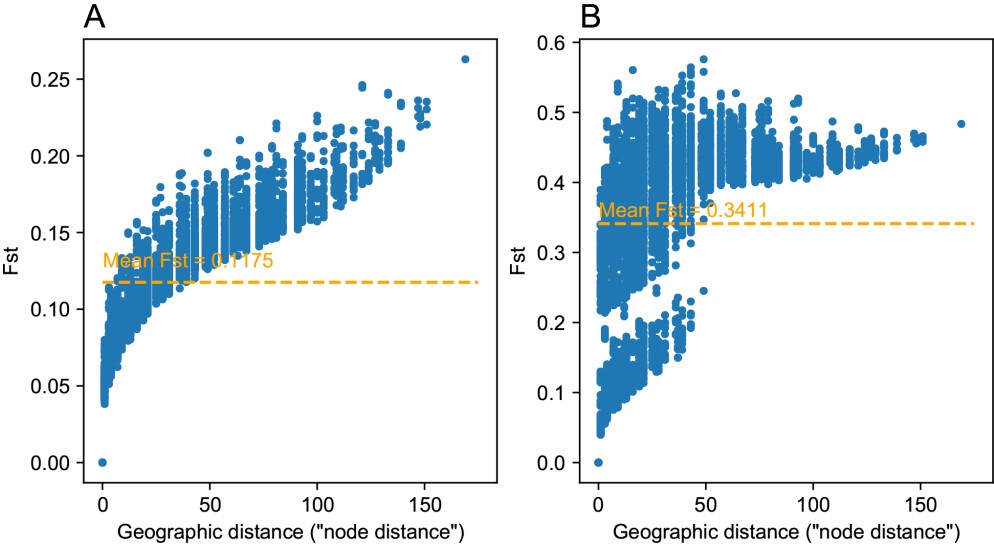

**Appendix 1—figure 20.** Comparing pairwise $F_{st}$ between strong and weak heterogeneous migration coalescent simulations: We visualize of the relationship between geographic distance and estimated pairwise $F_{st}$ (genetic distance) between nodes on the spatial grid for a weak heterogeneous migration simulation (**A**) and a strong heterogeneous migration (**B**). As expected the average $F_{st}$ is lower for the weak migration setting, and we observe a clear clustering like effect in the data for the strong heterogeneous migration simulation. This strong clustering effect can be attributed to pairwise comparisons of nodes across the region of reduced gene flow. Distances between nodes were set to one in the simulation, and so the units of geographic distance here are in units of the inter-node distance.

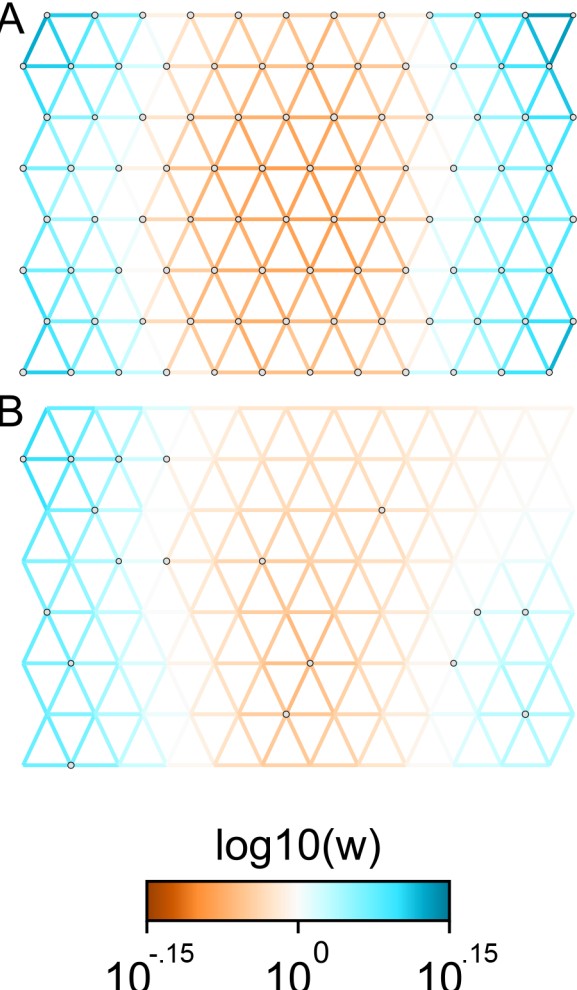

**Appendix 1—figure 21.** Applications of FEEMS to weak migration coalescent simulations: We visualize the FEEMS fit to coalescent simulations with weak heterogeneous migration. The coalescent migration rate in the center of the habitat is set to be 25% lower than the left or right regions. Note that the color-scale limits are set to $10^{-.15}$ and $10^{.15}$, respectively. The top panel shows the fit when all the nodes of the spatial graph are observed, whereas the bottom panel shows the fit when a sparse subset of nodes are observed. We see that FEEMS can still detect a signal of heterogeneity by displaying reduced gene-flow in the center of the habitat in the top panel. When we observe only a few nodes, in this weak migration setting, the FEEMS visualization looks overly smooth.

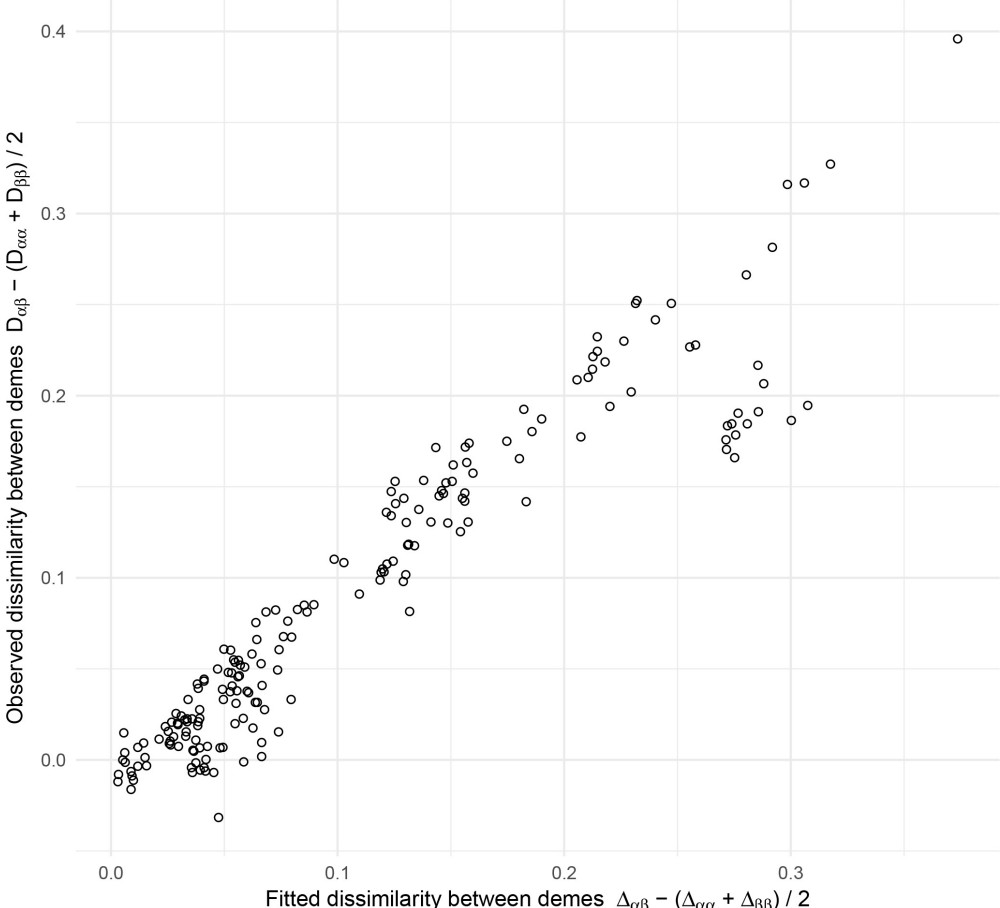

**Appendix 1—figure 22.** EEMS fitted versus observed genetic dissimilarities from the North American gray wolf dataset: We visualize fitted versus observed genetic dissimilarities corresponding to the EEMS visualization run on the North American gray wolf dataset in *Figure 4*. EEMS was run on a sparse grid with 307 nodes due to long run-times on the dense grid. Generally there is good concordance between the fitted and observed dissimilarities except for a small set of points whose fitted genetic dissimilarity over-estimates the observed dissimilarity, implying a relatively poorly fit for these points. Note, we do not see these poorly fit points in visualizations of the fitted versus observed distances when using FEEMS (see *Appendix 1—figure 6*).

