## [Decision Letter]

**Acceptance summary:**

The authors of the manuscript present a new implementation of the previously developed statistical method called "Estimating Effective Migration Surfaces", which displays on geographical map regions of low or high effective migration under a broad model of isolation by distance. In this new implementation migration surfaces are estimated under a penalized-likelihood approach coupled with optimization instead of MCMC leading to faster running times. The new implementation facilitates faster running times to make its usage computationally possible for a wider range of research groups and likely be applied to an even larger number of species/populations.

**Decision letter after peer review:**

Thank you for submitting your article "Fast and Flexible Estimation of Effective Migration Surfaces" for consideration by *eLife*. Your article has been reviewed by 2 peer reviewers, and the evaluation has been overseen by George Perry as the Senior and Reviewing Editor. The following individuals involved in review of your submission have agreed to reveal their identity: Isabel Alves (Reviewer #1); Wesley Tansey (Reviewer #2).

The reviewers have discussed the reviews with one another and the Reviewing Editor has drafted this decision to help you prepare a revised submission.

As the editors have judged that your manuscript is of interest, but as described below that additional analyses are required before it is published, we would like to draw your attention to changes in our revision policy that we have made in response to COVID-19 (https://elifesciences.org/articles/57162). First, because many researchers have temporarily lost access to the labs, we will give authors as much time as they need to submit revised manuscripts. We are also offering, if you choose, to post the manuscript to bioRxiv (if it is not already there) along with this decision letter and a formal designation that the manuscript is "in revision at *eLife*". Please let us know if you would like to pursue this option. (If your work is more suitable for medRxiv, you will need to post the preprint yourself, as the mechanisms for us to do so are still in development.)

Summary:

The authors of the manuscript present a new implementation of the previously developed statistical method called "Estimating Effective Migration Surfaces", which displays on a geographical map regions of low or high effective migration under a broad model of isolation by distance. In this new implementation migration surfaces are estimated under a penalized-likelihood approach coupled with optimization instead of MCMC leading to faster running times. The new implementation appears very promising as faster running times will make its usage computationally possible for a wider range of research groups and likely be applied to an even larger number of species/populations. Overall, we value the approach for its pragmatism but felt that it falls short at the very end by failing to provide any quantitative, objective way to choose the hyperparameters, which needs to be addressed as per the essential revisions detailed below.

Essential revisions:

1. The authors need to provide principled (or at least reproducible) ways to select the hyperparameters. Specific reviewer comments include:

"This is a major benefit of the L1 penalty. You can use BIC as the model selection criterion in the L1 case since the degrees of freedom are well-described. In the squared L2 case, it's not really possible. The authors discuss the issue and note LOOCV did not produce stable results, but they do not provide any data or examples. A more thorough investigation of hyperparameter settings is needed along with a recommendation that does not rely on biologists' subjective preference of the results on each dataset."

"FEEMS outcomes are very sensitive to user-based settings such as grid density and tuning parameters, as well as to aspects of the real data (eg. sampling design) that may result in an arbitrary choice of the outcome and lead to over-interpretations. I know the authors recommend to explore several combinations of regularization parameters and then compare FEEMS results with clustering/differentiation patterns based on approaches like ADMIXTURE or FST distances in order to support the results, nevertheless it is still difficult to grasp what's the best strategy to assess if a fitted graph is over-fitting the observed data or instead is pointing out to a real area of, let's say, low effective migration rate. Sentences like: "…, while setting up the tuning parameter ɑ to a value that we found that worked for multiple data applications.…" (lines: 225-226) or "it is helpful to look more closely at particular solutions that find balance between spatial homogeneity and complexity…" (lines: 243-245) are confusing and make difficult the choice of the final regularization parameters."

"The grid design is another arbitrary aspect of the method whose influence on the identification of regions of low or high migration isn't clear. Imagining one has the computational power to construct a very dense grid, is it worth doing it once there is observed data in 1% of the nodes? Is there a good relationship between density and number of sampled points? Does it affect the outcome?"

"I think the points above would be clearer if the authors would provide for instance, step-by-step guidelines to help future users of FEEMS and referring to specific examples in the manuscript in order to more clearly justify their parameter choice (eg ɑ = 50 line 226)."

2. Please clarify the modeling decision and its comparison to the L1 approach. For instance, isn't smoothing over nodes vs edges really just the same thing with different penalties? The authors form a lifted graph, where edges are now nodes and they penalize differences between neighboring edges. In the L1 penalty case with a squared error loss, the fused lasso / total variation penalty on neighboring edges is equivalent to linear trend filtering on the nodes. The choice then of linear trend filtering could have been replaced with a higher order trend filtering step to achieve the smoothness that the authors seem to say is lacking in the L1 model.

3. Are the data points we observe actually sampled at random? Is some sort of latent confounding likely? For example, maybe wolves migrate based on the season and the scientists collecting the data only look in one spot in one particular time of year?

4. Please clarify the simulation results in Supp Fig19, panels I and J. Without any data points in the orange regions for panel I, the model somehow infers that there is a band of different edge weights. How? In the 1d case, it's as if someone showed you: [5, 5, 5, missing, missing, missing, 5 ,5 5], and you come back and told me [5, 5, 5, -3, -3, -3, 5, 5, 5]. Is this possible?

5. At present, there is no real quantitative assessment of how good the FEEMS solutions are relative to the EEMS solution. This should be provided.

6. It would be useful to provide another example of a heterogeneous migration scenario where the reduction in migration is less than one order of magnitude in order to give an idea to the user of how the method performs in a less heterogeneous scenario (ie the lower bound).

---

## [Author Response]

Essential revisions:1. The authors need to provide principled (or at least reproducible) ways to select the hyperparameters. Specific reviewer comments include:"This is a major benefit of the L1 penalty. You can use BIC as the model selection criterion in the L1 case since the degrees of freedom are well-described. In the squared L2 case, it's not really possible. The authors discuss the issue and note LOOCV did not produce stable results, but they do not provide any data or examples. A more thorough investigation of hyperparameter settings is needed along with a recommendation that does not rely on biologists' subjective preference of the results on each dataset.""FEEMS outcomes are very sensitive to user-based settings such as grid density and tuning parameters, as well as to aspects of the real data (eg. sampling design) that may result in an arbitrary choice of the outcome and lead to over-interpretations. I know the authors recommend to explore several combinations of regularization parameters and then compare FEEMS results with clustering/differentiation patterns based on approaches like ADMIXTURE or FST distances in order to support the results, nevertheless it is still difficult to grasp what's the best strategy to assess if a fitted graph is over-fitting the observed data or instead is pointing out to a real area of, let's say, low effective migration rate. Sentences like: "…, while setting up the tuning parameter ɑ to a value that we found that worked for multiple data applications.…" (lines: 225-226) or "it is helpful to look more closely at particular solutions that find balance between spatial homogeneity and complexity…" (lines: 243-245) are confusing and make difficult the choice of the final regularization parameters.""The grid design is another arbitrary aspect of the method whose influence on the identification of regions of low or high migration isn't clear. Imagining one has the computational power to construct a very dense grid, is it worth doing it once there is observed data in 1% of the nodes? Is there a good relationship between density and number of sampled points? Does it affect the outcome?""I think the points above would be clearer if the authors would provide for instance, step-by-step guidelines to help future users of FEEMS and referring to specific examples in the manuscript in order to more clearly justify their parameter choice (eg ɑ = 50 line 226)."

We thank the reviewers for these helpful comments in regards to selecting the hyper-parameters of the penalty and agree that an automated selection procedure would help improve the interpretability and usability of FEEMS. To this end, we have made a number of updates to our modeling approach that have allowed for fully automated hyper-parameter selection and have proven to work well in practice. These developments were yielded through a simple but effective update of the parameterization of our penalty that allows for a cross-validation approach over just the smoothness parameter lambda alone. Specifically, we utilize the solution of the edge weights fitted under a homogenous migration model to penalize differences in neighboring edge weights on both the linear and log scale relative to a homogeneous fitted parameter which we call w_0_. This natural parameterization of the penalty and pre-estimation of w_0_ under a simple homogenous null model, allowed us to effectively remove the alpha parameter from the original penalty, allowing an one-dimensional cross-validation algorithm for selecting lambda in a computationally efficient and reliable manner. For more details please see the updated “Overview of FEEMS” in the Results section and “Penalty description” in the Materials and methods section.

With this new penalty in hand we used leave-one-out cross-validation to select the smoothness parameter lambda. In the cross-validation algorithm, for each grid value of lambda, we held-out an individual observed node (population) on the graph and then predicted underlying allele frequencies at these held-out nodes under our fitted spatial model from the rest of the training-set nodes (see the new section “Leave-one-out cross-validation to select tuning parameters” in Materials and methods for details). We found leave-one-out cross-validation over lambda to provide satisfactory results that recovered true migration histories in coalescent simulations and aligned with biological expectations in real datasets.

We have updated all of the text with descriptions of the new penalty and have reanalyzed and reproduced the figures using leave-one-out cross-validation to select lambda. This greatly simplifies the text and we hope this helps to alleviate the comments posed by the reviewers in regards to hyper-parameter selection. Figure 3 now shows fitted FEEMS visualizations across the grid points of lambda that were used in leave-one-out cross-validation. We added a new panel Figure 3E which shows the cross-validation error for the full grid and highlights the visualized maps for a subset of lambda values. In Figure 4, we now display the solution that achieves the minimum cross-validation error. All coalescent simulation figures have been updated as well. In general, the interpretation and results have not changed using this new penalty and cross-validation procedure but the ease of use and clarity of FEEMS has greatly improved.

We also thank the reviewers for the suggestion on using the BIC for model selection under the L1 penalty. As mentioned above, for our new penalty – which is still the L2 distance between migration weights on neighboring edges, we have found leave-one-out cross-validation to perform well. Because cross-validation, in principle, can work for both the L1 and L2 penalties we prefer to use a method that works more "universally" for any penalty whether it induces exact sparsity or not. We also prefer the L2 penalty for other statistical and computational reasons and expand upon this point in our response in the next section.

In terms of concerns about the grid density, we do not have new solutions for this problem which is also a caveat in the original EEMS method (see Petkova et al. 2016 discussion); however we highlight this issue more prominently with a new paragraph in the discussion.

2. Please clarify the modeling decision and its comparison to the L1 approach. For instance, isn't smoothing over nodes vs edges really just the same thing with different penalties? The authors form a lifted graph, where edges are now nodes and they penalize differences between neighboring edges. In the L1 penalty case with a squared error loss, the fused lasso / total variation penalty on neighboring edges is equivalent to linear trend filtering on the nodes. The choice then of linear trend filtering could have been replaced with a higher order trend filtering step to achieve the smoothness that the authors seem to say is lacking in the L1 model.

In a sense, yes, “smoothing over nodes vs edges is the same thing with different penalties”; however the penalties differ in key ways. In the original parameterization of EEMS, each node was given a parameter and the edge weights were deterministically computed as the average of adjacent connected nodes. While this node-level parameterization reduces the number of parameters needed to be estimated, the edge-level parameterization has two advantages in our view:

1. Each edge is free to take a unique value and that allows for a wider range of anisotropic migration scenarios to be modeled (e.g. spatially homogeneous anisotropy as in Figure 2 right hand column). That was the main driver for the decision for this new smoothing scheme.

2. By assigning each edge to be the average of connected nodes, a form of implicit spatial regularization is imposed because multiple edges connected to the same node would average that node’s parameter value. We found it more natural to separate out the regularization from the parameterization of the model. This preference led us to adding a smoothness penalty on edge-level parameters.

We thank the reviewer for the suggestion of using higher order trend filtering. We tested a L1 penalization approach on the edge weights and find it often fails to give satisfactory results (e.g. Supplementary Materials “Smooth penalty with L1 norm’’ and Supplementary Figure 18), primarily because the L1 penalty is too locally adaptive to regions with many unobserved locations. In the regime where there is a high degree of missingness in the graph, the global consideration of smoothing the unobserved locations seems to be necessary and this is the primary driver for using the L2 penalty. We believe that using the higher order trend filtering with the L1 penalty may suffer a similar issue. The smoothness of the L2 penalty also allowed us to employ a quasi-newton algorithm for optimizing our objective function which decreased our runtime more than 10 fold when compared to first order methods such as proximal gradient descent and ADMM when using the L1 formulation of our objective function. In addition we were able to utilize a widely used and tested implementation of L-BFGS in scipy which worked well out of the box and had few algorithmic parameters to tune. Given we observed satisfactory results with the L2 penalty in addition to the fast convergence and runtime of the quasi-newton algorithm we preferred it over the L1 penalization approach.

3. Are the data points we observe actually sampled at random? Is some sort of latent confounding likely? For example, maybe wolves migrate based on the season and the scientists collecting the data only look in one spot in one particular time of year?

For (a), in our model we treat the geographic locations of each sample as fixed but the distribution of genetic data as random. We find the method is relatively robust to non-random sampling, as long as it is not so sparse as to lose the key signals of differentiation in the data (e.g. Supplementary Figure 2).

Regarding (b), we have expanded the discussion with a paragraph that addresses how a form of confounding between seasonal migration and the sample collection process could be problematic in some datasets. Specifically, we discuss variation in the wolf migratory behavior to illustrate the point, and the suggestion helped us add nuance to our discussion of the results.

4. Please clarify the simulation results in Supp Fig19, panels I and J. Without any data points in the orange regions for panel I, the model somehow infers that there is a band of different edge weights. How? In the 1d case, it's as if someone showed you: [5, 5, 5, missing, missing, missing, 5 ,5 5], and you come back and told me [5, 5, 5, -3, -3, -3, 5, 5, 5]. Is this possible?

It is possible – and there are a few ways to understand where the signal for the inference derives from. (1) Consider that in sampled regions one can “learn” a relationship between geographic and genetic distance, and then with that in hand recognize that the observed covariances across a gap of unobserved locations are too large or too small relative to what is seen in the observed data. Regarding the reviewer’s analogy, let’s define a spatial position of each node as its index in the 1-D array. Let's further suppose we collected a dataset of samples from nodes 1,2,3 and 7,8,9. First, assume we see that the observed allele frequency covariance within pairs of nodes 1,2,3 and 7,8,9 decays some constant amount for every unit of geographic distance. Then, if the covariance between pairs nodes 1,2,3 and 7,8,9 is lower than what would be expected given the geographic distance between them, it would provide a signal that the migration rates should be higher for edges connecting 1,2,3 and 7,8,9 and lower for edges connecting 4,5,6. In contrast, if pairwise covariances between all of the observed nodes decayed over geographic space at a constant level, we would infer homogenous migration rates for edges connecting both the missing and non-missing nodes. Our likelihood uses the migration rate parameters to match expected covariances at all nodes to observed covariances, whereas our penalty encourages the migration rates to be smooth which helps inference in regions with high levels of missing nodes. As a simple one-locus example, consider the allele frequency data observed at the same 1-D array of 9 populations: [0.02,0.01,0.03, missing, missing, missing, 0.99,0.98,0.985]. From such allele frequency data, one can infer migration rates are unlikely to be homogeneous, and moreover, likely are lower in the region with no observed data. (2) In a population genetic model, the key determinant of variation in genetic data observed today is where and when the ancestors of the sampled data “coalesce” (i.e. have common ancestry). The genetic ancestors of the present sample can occupy locations where there is no data today, and when they coalesce with each other will be impacted by local migration rates at unsampled locations. That is to say, migration rates at locations where there are no samples today can still impact the genetic data observed. Surprisingly, this implies that, to some extent, one can learn about migration rates even outside the convex hull of the sampled points.

5. At present, there is no real quantitative assessment of how good the FEEMS solutions are relative to the EEMS solution. This should be provided.

We now include a new supplemental figure 22 with the observed vs fitted dissimilarities output by EEMS when applied to the North American gray wolf dataset. This can be compared to the analogous results for FEEMS already presented in supplemental figure 14. The results show that EEMS fits the data relatively well, though there is a collection of points that seem to be poorly fit as illustrated by a systematically higher fitted dissimilarity to what is observed. FEEMS, in contrast, does not seem to have the same cluster of poor fitting samples.

6. It would be useful to provide another example of a heterogeneous migration scenario where the reduction in migration is less than one order of magnitude in order to give an idea to the user of how the method performs in a less heterogeneous scenario (ie the lower bound).

We now include such an example. To assess the performance of FEEMS in a less heterogeneous migration scenario, we applied FEEMS to coalescent simulations where the migration in the center of the habitat (spatial grid) was only reduced by 25% relative to the edges. In Supplementary Figure 21, we can see that FEEMS is still able to recover this reduction of gene-flow but the output visualization is slightly noisier than the strong heterogeneous migration simulations in Figure 2. Please refer to a new supplementary section titled “Coalescent simulations with weak migration” for a detailed discussion of the results.